# Spermine synthase deficiency causes lysosomal dysfunction and oxidative stress in models of Snyder-Robinson syndrome

Chong Li [1], Jennifer M. Brazill[1], Sha Liu[2], Christofer Bello[1], Yi Zhu[1], Marie Morimoto[3,4], Lauren Cascio[5], Rini Pauly[5], Zoraida Diaz-Perez[1], May Christine V. Malicdan [3,4,6], Hongbo Wang[2], Luigi Boccuto[5], Charles E. Schwartz[5], William A. Gahl[3,4,6], Cornelius F. Boerkoel[3,4,7] & R. Grace Zhai [1,2]

Polyamines are tightly regulated polycations that are essential for life. Loss-of-function mutations in spermine synthase (SMS), a polyamine biosynthesis enzyme, cause Snyder-Robinson syndrome (SRS), an X-linked intellectual disability syndrome; however, little is known about the neuropathogenesis of the disease. Here we show that loss of *dSms* in *Drosophila* recapitulates the pathological polyamine imbalance of SRS and causes survival defects and synaptic degeneration. SMS deficiency leads to excessive spermidine catabolism, which generates toxic metabolites that cause lysosomal defects and oxidative stress. Consequently, autophagy–lysosome flux and mitochondrial function are compromised in the *Drosophila* nervous system and SRS patient cells. Importantly, oxidative stress caused by loss of *SMS* is suppressed by genetically or pharmacologically enhanced antioxidant activity. Our findings uncover some of the mechanisms underlying the pathological consequences of abnormal polyamine metabolism in the nervous system and may provide potential therapeutic targets for treating SRS and other polyamine-associated neurological disorders.

[1] Department of Molecular and Cellular Pharmacology, University of Miami Miller School of Medicine, Miami, FL 33136, USA. [2] School of Pharmacy, Key Laboratory of Molecular Pharmacology and Drug Evaluation (Yantai University), Ministry of Education, Collaborative Innovation Center of Advanced Drug Delivery System and Biotech Drugs in Universities of Shandong, Yantai University, Yantai, Shandong 264005, China. [3] NIH Undiagnosed Diseases Program, National Human Genome Research Institute, NIH, Bethesda, MD 20892, USA. [4] Section of Human Biochemical Genetics, Medical Genetics Branch, National Human Genome Research Institute, NIH, Bethesda, MD 20892, USA. [5] JC Self Research Institute, Greenwood Genetic Center, Greenwood, SC 29646, USA. [6] Office of the Clinical Director, National Human Genome Research Institute, NIH, Bethesda, MD 20892, USA. [7] Present address: Department of Medical Genetics, University of British Columbia, Vancouver, BC V6H 3N1, Canada. Correspondence and requests for materials should be addressed to R.G.Z. (email: gzhai@med.miami.edu)

Polyamines, including putrescine, spermidine, and spermine, participate in various cellular events including chromatin structure modulation and transcriptional and translational regulation and are critical for normal cellular physiology[1,2]. Polyamine homeostasis is maintained in a cell-type-specific manner and is tightly regulated through de novo synthesis, inter-conversion, and transportation[3]. Altered polyamine metabolism has been associated with aging and various diseases, including cancer, inflammation, and neurological disorders[4–8]. The particular importance of polyamine homeostasis in the nervous system is highlighted by the altered polyamine metabolism detected in stroke and brain injury[6,7,9]. Additionally, restoring polyamine levels by dietary supplementation has recently been reported to protect against age-induced memory impairment and to modulate circadian periods[10,11]. Although the importance of polyamine balance is increasingly recognized, the in vivo mechanisms underlying the detrimental consequences of metabolic disruption and polyamine imbalance remain unclear.

The polyamine biosynthesis pathway consists of two successive steps catalyzed by two aminopropyltransferases: spermidine synthase converts putrescine to spermidine, and spermine synthase (SMS) converts spermidine to spermine[12,13]. In the past decade, mutations in human SMS (hSMS) have been found to cause the X-linked intellectual disability (XLID) Snyder-Robinson syndrome (SRS, OMIM 309583)[14–18]. SRS is characterized by a collection of clinical features including mild to severe intellectual disability, hypotonia, skeletal defects, movement disorders, speech and vision impairment, seizure, and cerebellar circuitry dysfunction[15,18,19]. SRS was one of the earliest reported XLID syndromes[19] and thus far the only known genetic disorder associated with the polyamine metabolic pathway. So far, animal models for studying the pathology of SRS, as well as polyamine-associated neurological disorders, are limited. The hemizygous Gy male mice with partial deletion of both SMS and downstream gene PHEX (phosphate-regulating endopeptidase homolog, X-linked) was originally used as a model for X-linked hypophosphatemia for defects in phosphate transport[20,21]. Gy mice have decreased spermine levels and in addition to hypophosphatemic rickets show neurological phenotypes including circling behavior and inner ear abnormalities; however, the compounding effects from loss of PHEX function made it difficult to pinpoint the underlying pathophysiology of SMS deficiency[22].

Here we establish a Drosophila model for SRS, where Drosophila Sms (dSms) mutant flies recapitulate the pathological spermidine accumulation in patients with SRS. Our mechanistic analysis of SMS-deficient Drosophila nervous system and SRS patient cells show that altered polyamine metabolism leads to abnormal spermidine catabolism and accumulation of toxic metabolites that compromise lysosomes, disrupt autophagic–lysosomal flux, produce oxidative stress, and impair mitochondrial function. Using metabolic phenotypic microarray, we analyze the metabolic profile of SRS patient cells and uncover distinct signatures of SRS that may facilitate diagnosis. Importantly, our findings reveal targets for genetic and pharmacological antioxidant defenses with promising therapeutic potential for SRS and other polyamine-associated neurological disorders.

## Results

### CG4300 is the Drosophila homolog of human SMS.
SMS catalyzes transfer of the aminopropyl group of decarboxylated S-adenosylmethionine (dcAdoMet) onto spermidine to form spermine and a byproduct, methylthioadenosine (MTA)[23]. To date, 14 mutations in the SMS gene causing SRS have been identified, 4 of which are first reported here[14–17] (Fig. 1a).

Drosophila has one predicted SMS gene (CG4300; hereafter referred to as dSms) with two annotated transcripts, dSms RA and RB, producing two isoforms, PA and PB, with identical catalytic and binding residues (Fig. 1b and Supplementary Fig. 1). SMS is conserved across distant phyla; Drosophila (dSMS) and human (hSMS) orthologues share 43% identity and 61% similarity in amino acid sequence (Fig. 1c). To establish a Drosophila model with dSms deficiency, we obtained an allele with a transposable element inserted in the intron between exon 3 and 4 (dSms$^e$, Fig. 1b). Comparative quantitative reverse transcriptase PCR (qRT-PCR) analysis indicated homozygous (dSms$^{e/e}$) and heterozygous (dSms$^{e/+}$) mutant flies have dSms transcript levels of less than 0.05% and ~50%, respectively, of control (yw) flies (Fig. 1d). dSms$^{e/e}$ flies have reduced viability (Fig. 1e) that is rescued by ubiquitous expression (actin-GAL4) of Drosophila (UAS-dSms$^{RA}$) and human (UAS-hSMS$^{wt}$) wild-type SMS. The mutation c.443A>G (hSMS$^{443}$, p.Gln148Arg), which affects the highly conserved 5′-MTA-binding site, was reported most recently by the NIH Undiagnosed Diseases Program (UDP) in two brothers with severe and expanded SRS phenotypes[14] (Fig. 1a). Expression of human mutant SMS (UAS-hSMS$^{443}$) did not rescue viability as well as hSMS$^{wt}$ (Fig. 1e). These results confirmed functional homology of dSms$^{RA}$ and hSMS$^{wt}$ and the pathogenicity of hSMS$^{443}$.

All identified mutations of SRS cause partial to complete loss of SMS enzyme activity and a consequent block in the polyamine synthesis pathway[13–18] (Fig. 1f). Consistently, SRS patients have increased spermidine levels[14–18]. We determined the polyamine levels in dSms$^{e/e}$ flies by liquid chromatography tandem mass spectrometry (LC-MS/MS) and found that spermidine levels were significantly increased in both male and female dSms$^{e/e}$ flies (Fig. 1g). The level of the upstream polyamine putrescine was decreased in male dSms mutant flies (Fig. 1g), consistent with several reports of decreased putrescine levels in SRS patients[15,16]. In addition to attenuated viability and spermidine accumulation, dSms$^{e/e}$ flies experienced shortened lifespan and severe locomotor deficits suggestive of nervous and/or muscular system dysfunction (Fig. 1h, i). dSms$^{RA}$ overexpression rescued the lifespan and significantly restored the locomotor behavior (Fig. 1h, i), suggesting that dSms$^{RA}$ is a functional isoform and the defects observed in dSms$^{e/e}$ are caused by loss of dSms.

### Loss of dSms causes retinal and synaptic degeneration.
To explore the role of dSms in the central nervous system, we examined the expression profile and found dSms transcript is expressed during all developmental stages (Supplementary Fig. 2a, b). Using a GFP trap line (dSms-GFP$^{CB04249}$), we found dSms protein is ubiquitously present in the nervous system (Supplementary Fig. 2c–h). To establish the nervous system requirement of dSms, we focused on the visual system where we found abnormal retinal vacuoles in dSms$^{e/e}$ flies at 2 days after eclosion (DAE) (Supplementary Fig. 3). Additionally, we observed an age-dependent (Fig. 2c–e) loss of pigmentation in dSms$^{e/e}$ flies characteristic of retinal degeneration[24]. Retinal pigmentary changes have also been reported in SRS patients[14]. Consistent with retinal degeneration, dSms$^{e/e}$ flies exhibited age-dependent reduction of photoreceptor depolarization and synaptic response shown by the on/off transients, which can be rescued by overexpressing dSms$^{RA}$ (Fig. 2a′–f′, quantified in 2g–i). These results indicate early onset and progressive neuronal dysfunction in dSms mutant visual system.

Transmission electron microscopy (TEM) analysis of the dSms$^{e/e}$ visual synapse revealed a disorganized synapse within the lamina, the first neuropil of the fly visual system (Fig. 2j, k).

We also observed two pervasive morphological irregularities. First, single layer (type I), double layer (type II), and multilayer membrane structures (type III), as well as multilamellar bodies (type IV) accumulated in the presynaptic photoreceptor terminals (Fig. 2l, m) suggesting defects in membrane compartment dynamics at the presynaptic site. These membrane structures were seldom present in wild-type synapses. Second,

ultrastructural analysis revealed a decreased number and enlarged area of presynaptic but not post-synaptic mitochondria (Fig. 2n, o); a higher percentage of mutant presynaptic mitochondria was also without cristae (Fig. 2o). These results indicate that presynaptic terminals are highly sensitive to loss of dSms, likely owing to the high energy and membrane trafficking demands of this compartment.

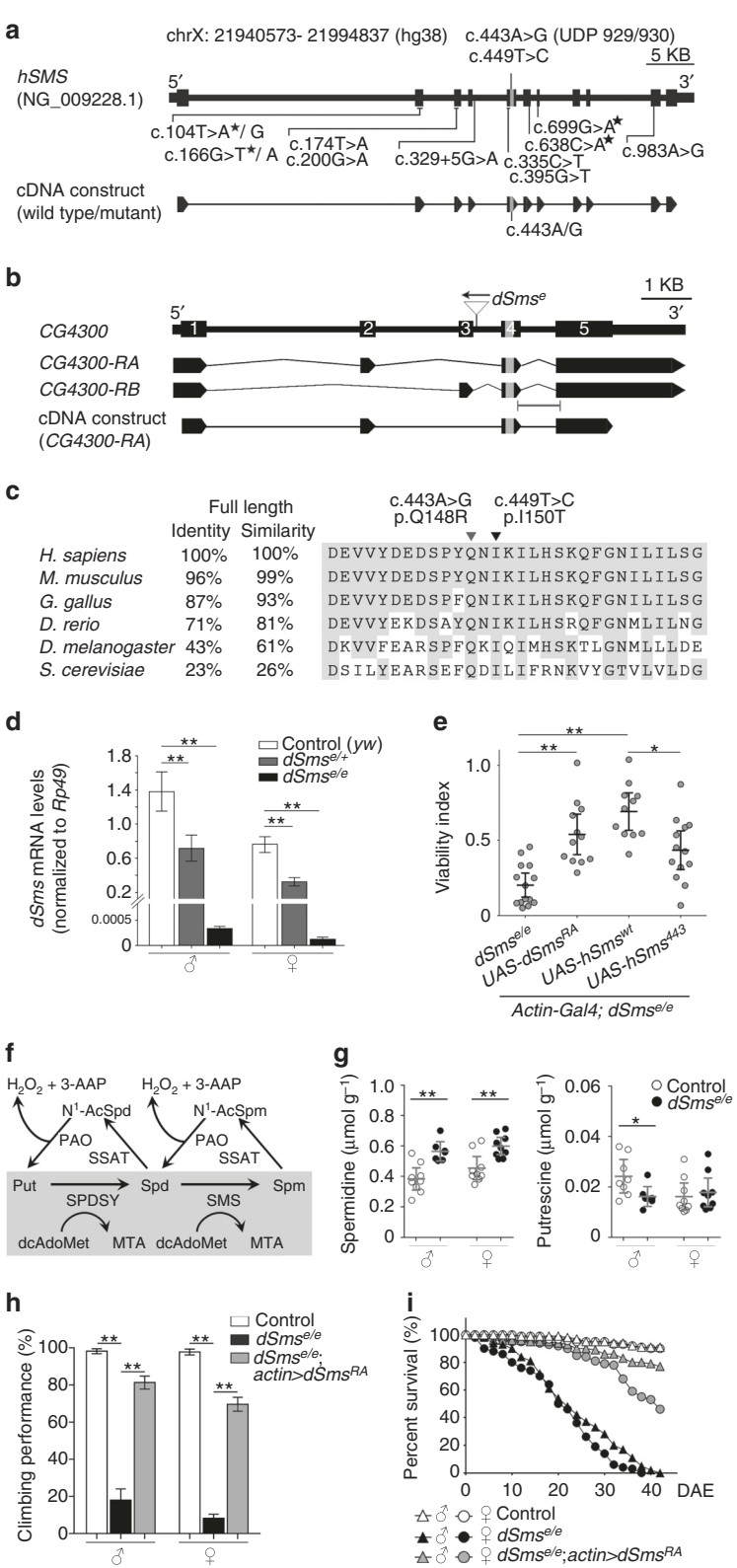

**SMS deficiency impairs autophagy.** To delineate the membrane abnormalities in dSms mutant synapses, we examined membrane trafficking, including endosomal sorting and autophagy–lysosome flux. Disruption of these pathways has been linked to abnormal membrane accumulation and neuronal dysfunction[25]. We used immunofluorescence to measure *Drosophila* Rab5, an early endosomes marker[26]; Atg8a, an Atg8/LC3 protein family member and a marker for autophagosomes[27]; and Ref(2)P, the *Drosophila* homolog of human p62/SQSTM1, a marker for ubiquitinated protein aggregates degraded through autophagy[28]. $dSms^{e/e}$ mutant flies exhibited increased Rab5 intensity in lamina synapses (Fig. 3b and Supplementary Fig. 4a), as well as a drastic increase in Atg8- and Ref(2)P-positive puncta in the cell body layer of lamina neurons (Fig. 3c, d).

The accumulation of endosomes and autophagosomes suggests either an activation of the endosome–autophagy pathway or a block of downstream autophagy flux. To differentiate between these possibilities and visualize the downstream autophagosome–lysosome flux, we used an mCherry-GFP-Atg8a reporter with a pH-sensitive GFP[29]. The GFP fluorescence is quenched upon fusion of the autophagosome with the acidified lysosome, whereas fluorescence from acid-insensitive mCherry remains until the protein is degraded[29] (Fig. 3e). We expressed this reporter in the wild type and dSms mutant flies and examined the fluorescence intensity of the puncta in the lamina (Fig. 3f, g). Puncta with high mCherry and high GFP signals (Fig. 3h, quadrant I) indicate either Atg8a-positive autophagosomes or dysfunctional autolysosomes with a neutral pH, since fusion of autophagosome and lysosome can proceed independent of lysosome acidification[30]. Puncta with high mCherry and low GFP signals (Fig. 3h, quadrant II) indicate fusion with the acidic lysosome. Puncta with low mCherry and low GFP signals (Fig. 3h, quadrant III) indicate normal lysosomal degradation of Atg8a. With this classification system, we found that within the control lamina 25% of Atg8 puncta-labeled autophagosomes (I), about 25% labeled autophagosomes undergoing fusion with the lysosome (II), and more than 50% were undergoing degradation within the lysosome (III); this defined basal autophagic flux (Fig. 3f, h). $dSms^{e/e}$ lamina, however, had significantly more Atg8 puncta associated with autophagosomes or dysfunctional autolysosomes (I) and fewer puncta incorporated into normal autolysosomes (II and III) as well as increased GFP intensity throughout the lamina (Fig. 3f–h and Supplementary Fig. 4b).

In parallel with the mCherry-GFP-Atg8a assay in flies, autophagic flux was monitored in cultured patient cells by viral transduction of an RFP-GFP-LC3 reporter. We observed no difference in autophagic flux in fibroblasts from SRS patients, however we observed significantly more autophagosomes or dysfunctional autolysosomes along with fewer acidic

autolysosomes in bone marrow stromal cells (BMSCs) from SRS patients, a cell type that exhibits a robust molecular phenotype with respect to polyamine levels compared to SRS fibroblasts[14] (Fig. 4a, b and Supplementary Fig. 5a, b). Lysosomal turnover of endogenous LC3-phospholipid conjugate (LC3-II) in the conditions with/without autophagy inhibition or induction has been used to examine autophagic flux[31]. We next performed biochemical analysis of autophagic markers from patient cells subjected to a series of conditions to manipulate autophagy. Consistent with the immunofluorescent data, the SRS fibroblasts show unaltered LC3-II levels under basal conditions (untreated) compared with control (Supplementary Fig. 5c, d), while SRS BMSCs show decreased LC3-II levels under basal conditions (untreated) compared with control (Fig. 4c, d and Supplementary Fig. 6a). As expected, when autophagy was either inhibited (bafilomycin) or induced (starvation), control and SRS BMSCs showed increased LC3-II levels (Fig. 4c, d). In both conditions, lower levels of LC3-II were detected in SRS BMSCs (Fig. 4c, d), which suggest defects in LC3-II turnover. However, no significant differences of p62 levels between control and SRS were observed in BMSCs or fibroblasts, even when autophagy was inhibited or induced (Fig. 4c, e and Supplementary Fig. 5e, f), which may be due to the selective target of p62[28].

Together, these results suggest that SMS deficiency leads to the accumulation of autophagosomes through defects in autophagosome–lysosome fusion and/or lysosomal degradation. The cellular phenotypes may be more pronounced in specific cell types, including BMSCs and neurons.

**Abnormal spermidine catabolism compromises lysosome function.** Reactive aldehydes generated during polyamine catabolism are highly cytotoxic[32–35]. As elevated spermidine levels in $dSms^{e/e}$ flies are a potential source for such reactive aldehydes, we examined spermidine catabolism by (1) determining the level of $N^1$-acetylspermidine, the limiting derivative in polyamine catabolism[36,37] using LC-MS/MS and (2) measuring the total aldehyde content using a colorimetric method[38]. Both $N^1$-acetylspermidine and total aldehyde levels were increased in $dSms^{e/e}$ flies (Fig. 5a, b). Higher acetylspermidine levels have also been reported in several SRS patients by clinical metabolomics[39].

Multilamellar membranous structures similar to those in the dSms mutant synapses (Fig. 2l, type IV) have been linked to lysosomal dysfunction in both *Drosophila* and mammalian models of neurodegeneration[40–42]. Several reactive aldehydes are specifically lysosomotropic[32,43], so we hypothesized that lysosomal activity is impaired by the polyamine imbalance observed in $dSms^{e/e}$ *Drosophila* nervous system and SRS patient cells. Immunohistochemical analysis revealed diffuse lysosome-

**Fig. 1** Loss of *dSms* recapitulates polyamine imbalance of SRS and causes survival defects in *Drosophila*. **a** Diagram of human *SMS* gene structure and wild-type/mutant cDNA constructs. Fourteen mutations identified in SRS are indicated. The asterisks indicate new mutations. **b** Diagram of *Drosophila CG4300* gene structure, predicated mRNA transcripts, and cDNA construct. $dSms^e$ allele has a *P* element inserted in the opposite direction of the gene (triangle). Capped line under mRNAs indicates qPCR probe. **c** Homology of SMS proteins across species scored by amino acid identity and similarity. Protein sequence (coded by gray region indicated in **a** and **b**) alignment showing evolutionary conservation (gray background). Arrowheads indicate amino acids affected in SRS. **d** qPCR of *dSms* normalized to *Rp49* in control, $dSms^{e/+}$, and $dSms^{e/e}$ fly heads (mean ± S.E.M.; $n = 4$ extractions, each extraction from 10 fly heads). **e** Scatter dot plot of viability index (mean ± 95% CI, $n = 14$ for $dSms^{e/e}$, 12 for $dSms^{RA}$ rescue, 11 for $hSms^{wt}$ rescue, and 13 for $hSms^{443}$ rescue; each data point represents a sample of ≥80 embryos) of control, $dSms^{e/e}$, and $dSms^{e/e}$ flies ubiquitously expressing cDNA constructs. **f** Diagram of polyamine metabolic pathway (Put putrescine, Spd spermidine, Spm spermine, SPDSY spermidine synthase, $N^1$-AcSpd $N^1$-acetylspermidine, PAO polyamine oxidase, SSAT spermidine/spermine acetyltransferase, 3-AAP 3-acetoamidopropanal). **g** Absolute levels (mean ± 95% CI, $n = 9/7$ for male control/$dSms^{e/e}$ and 9/9 for female control/$dSms^{e/e}$ spermidine levels; $n = 8/7$ for male control/$dSms^{e/e}$ and 9/9 for female control/$dSms^{e/e}$ putrescine levels; each data point represents a sample of ≥20 flies) of polyamines in the whole body of young (2–5 DAE) flies measured by LC-MS/MS. **h** Climbing performance (mean ± S.E.M.; each data point obtained from a group of 10 individuals, $n = 10$ experiments) of male and female flies at 2 DAE. **i** Fly survival curves ($n = 109/106$ for control male/female, 75/62 for $dSms^{e/e}$ male/female, and 130/132 for $dSms^{e/e}$; *actin* > $dSms^{RA}$ male/female flies) determined by the age-specific number of live individuals. **d**, **e**, **h**, One-way ANOVA post hoc Bonferroni test (**d**) or Tukey test (**e**, **h**), and **g** Student $t$ test; *$P < 0.05$, **$P < 0.01$

associated membrane protein 1 (LAMP1) and reduced lysosomal-resident protease cathepsin L (CtsL) in $dSms^{e/e}$ synapses (Fig. 5c, d). Maturation of the cathepsin family of proteases requires transport into the late endosome/lysosome where the acidic environment promotes cleavage of the propeptide to generate the active form[44]. We performed western analysis of $dSms^{e/e}$ brain extracts and detected increased levels of pro-CstL

and decreased amounts of mature CtsL, indicating a deficit in lysosomal processing (Fig. 5e, f and Supplementary Fig. 6b).

Next we analyzed lysosomal function in cultured SRS patient dermal fibroblasts. Double immunostaining for LAMP1 and cathepsin D (CtsD) showed a LAMP1-positive ring structure often surrounding CtsD in control human fibroblasts (Fig. 5g), whereas fibroblasts from the SRS patients presented two

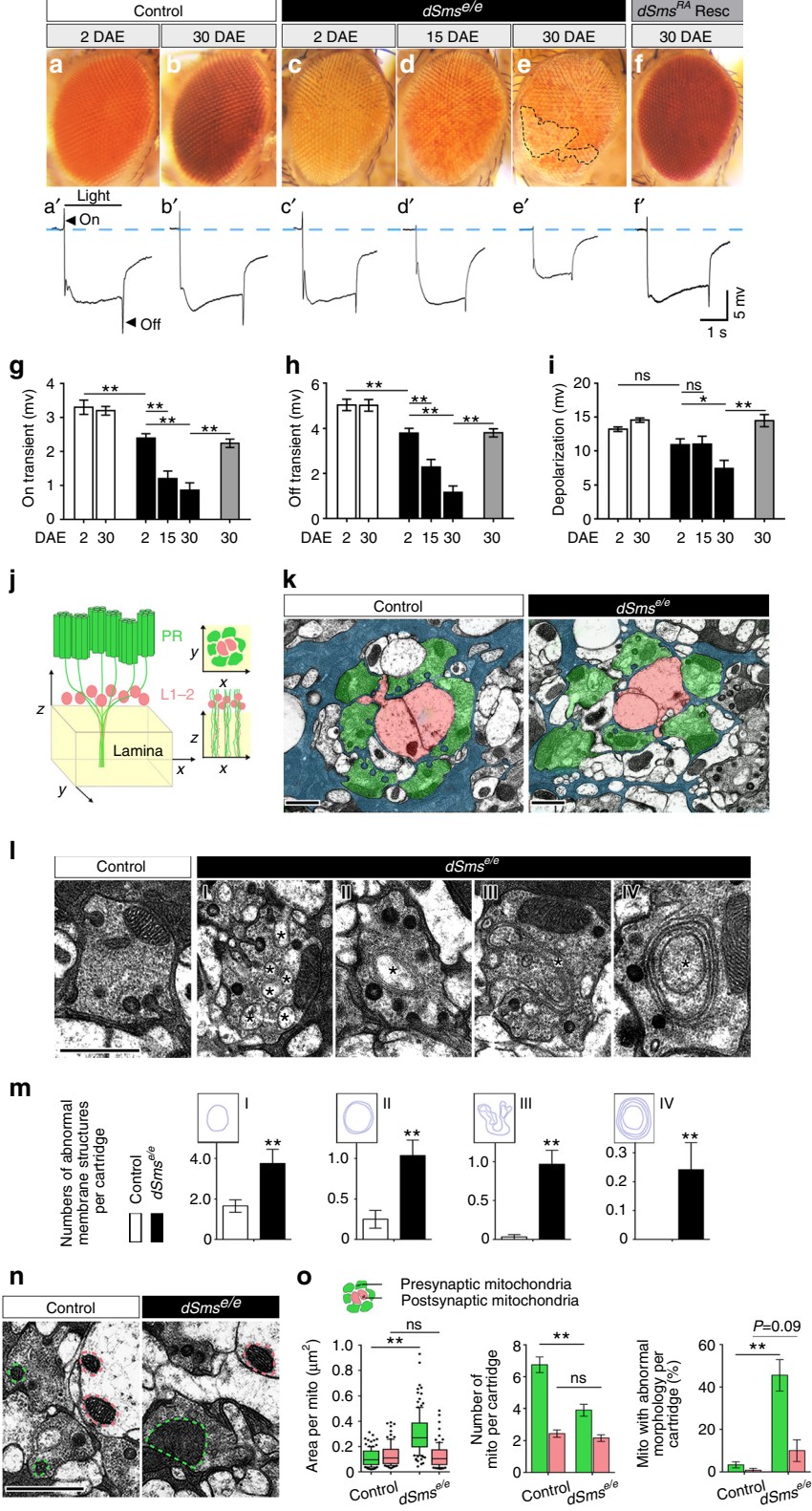

lysosomal phenotypes: (1) enlarged LAMP1-positive lysosomes lacking luminal CtsD and (2) small lysosomes with a dense center containing limited CtsD (Fig. 5g and Supplementary Fig. 7a–e). Biochemical analysis also detected decreased amounts of mature CtsD in SRS patient fibroblasts (Fig. 5h, i and Supplementary Fig. 6c). To further examine the integrity and specifically the luminal pH of lysosomes, we performed live imaging assays using LysoSensor yellow/blue and LysoTracker red probes on human fibroblasts[45,46]. A decreased yellow fluorescent was observed in SRS fibroblasts using LysoSensor dye, which clearly suggests a higher pH of the lysosomes (Fig. 5j). Additionally, while control fibroblasts show readily labeled acidic organelles with Lyso-Tracker, SRS fibroblasts show a diffuse LysoTracker dye labeling (Fig. 5k), indicating unhealthy lysosomes.

Collectively, these data suggest that in the intact *Drosophila* nervous system and cultured SRS patient fibroblasts, SMS deficiency leads to spermidine accumulation and subsequent acetylation and oxidation that lead to compromised lysosomal integrity and function.

**SMS deficiency impairs mitochondrial function.** To uncover cellular metabolic dysfunction in SRS patient cells, we carried out a screen on cultured lymphoblastoid cells using the Phenotype MicroArray (PM) platform. Compared to that of control, the metabolic profile of 14 unrelated SRS cell lines had a distinctive shift in metabolism of carbon energy sources (Fig. 6a and Supplementary Fig. 8a). Specifically, sugars—dextrin, glycogen, maltotriose, maltose, and D-fructose—were overutilized (Fig. 6b), whereas compounds metabolized in the Krebs cycle—citric acid, methyl pyruvate, α-ketoglutaric acid, succinic acid, L-malic acid, and D-malic acid—were underutilized (Fig. 6c). These results suggest that lymphoblastoid cells of SRS patients preferentially generate energy via glycolysis, rather than the Krebs cycle, implying reduced mitochondrial function. This SRS signature metabolic profile is unique compared to mutations in other genetic causes of intellectual disabilities such as autism (Supplementary Fig. 8b, c) and potentially provides a tool to facilitate diagnosis.

The reduced mitochondrial metabolism implicated in SRS patient cells is compatible with the morphologically abnormal mitochondria observed in $dSms^{e/e}$ photoreceptor terminals (Fig. 2n, o). To assess mitochondrial energy production in $dSms^{e/e}$ fly heads, we measured ATP at different ages with a bioluminescence assay and detected a decline in ATP content relative to control (Fig. 6d). We employed histochemical analysis to examine the activity of cytochrome c oxidase (COX, complex IV), the last enzyme of the electron transport chain, which revealed lower COX activity in brains and flight muscles of $dSms^{e/e}$ flies (Fig. 6e) suggesting reduced mitochondrial

respiration as the origin of diminished ATP content. The reduction in COX activity was restored by $dSms^{RA}$ expression, suggesting the COX defects were due to loss of *dSms* (Fig. 6e). Our findings reveal a perturbation of mitochondrial metabolism in both SRS patient cells and *dSms* mutant flies.

**Polyamine oxidation causes oxidative stress in fly brains.** Compromised mitochondrial function is often linked with alterations in oxidative state of the cell. Besides reactive aldehydes, another byproduct of spermidine catabolism is the reactive oxygen species (ROS) hydrogen peroxide ($H_2O_2$)[36]. We hypothesized that ROS from polyamine imbalance in $dSms^{e/e}$ flies causes oxidative stress, which can impair mitochondria function. Staining with superoxide sensitive dihydroethidium (DHE) indeed detected elevated ROS levels in the mutant brain (Fig. 7a, j). To confirm the deleterious effect of ROS accumulation, we boosted antioxidant defenses by overexpressing *Drosophila* glutathione S-transferase E1 (GstE1), a homolog of the human θ-class GSTs enzymes. GSTs conjugate reduced glutathione to toxic reactive compounds to promote cellular resistance to oxidative stress[47]. Both ubiquitous (*actin-GAL4*) and neuronal-specific (*elav-GAL4*) expression of GstE1 alleviated ROS overload in $dSms^{e/e}$ brains (Fig. 7b, j). Ameliorating ROS burden of $dSms^{e/e}$ flies by ubiquitous expression of GstE1 also increased viability, prolonged lifespan, and partially rescued the climbing deficits (Supplementary Fig. 9 and Fig. 7c, d). At the cellular level, GstE1 expression prevented neuronal degeneration in the visual system, including loss of pigment in the exterior eye and defects in phototransduction (Supplementary Fig. 10 and Fig. 7e–h), and improved function of mitochondria, more robustly in the flight muscle (Fig. 7k and Supplementary Fig. 11).

**ROS can be pharmacologically suppressed.** We next tested pharmacological approaches suitable for therapeutic intervention. We supplemented fly food with the antioxidants N-acetylcysteine amide (AD4) or N-2-mercaptopropionil glycine (N-2-MPG)[48,49] and found that AD4 ($40\,\mu g\,ml^{-1}$) and N-2-MPG (80 and $160\,\mu g\,ml^{-1}$) reduced ROS burden in the brains of $dSms^{e/e}$ flies (Fig. 7i, j). In addition, AD4 feeding increased COX activity, suggesting partial restoration of mitochondrial function (Fig. 7k and Supplementary Fig. 11). We also examined the effects of antioxidant feeding on survival. dSms mutant flies have a reduced survival rate, that is less than 50% of the flies complete meta-morphosis and eclosion process (Fig. 1e). Feeding dSms mutant larvae with antioxidant AD4 ($40\,\mu g\,ml^{-1}$) did not significantly improve viability (Supplementary Fig. 9), suggesting that feeding for 3 days at larval stage is insufficient to overcome metamorphosis defects and improve eclosion rate. Furthermore, administration of AD4 ($40\,\mu g\,ml^{-1}$) or N-2-MPG ($160\,\mu g\,ml^{-1}$) did not

**Fig. 2** Retinal degeneration and synaptic dysfunction present in $dSms^{e/e}$ flies. **a–f** Eye exterior morphology of flies at different ages. Note the loss of pigments of 30 DAE $dSms^{e/e}$ flies (**e** black dashed outline). **a′–f′** Representative traces from ERG recordings showing light-induced depolarization and on/off responses (black arrow heads). **g–i** Quantification (mean ± S.E.M.; n = 11 (control, 2 DAE), 10 (control, 30 DAE), 14 ($dSms^{e/e}$, 2 DAE), 10 ($dSms^{e/e}$, 15 DAE), 10 ($dSms^{e/e}$, 30 DAE), 10 ($dSms^{RA}$ rescue, 30 DAE) field recordings from four animals) of on transient (**g**), off transient (**h**), and depolarization (**i**) of ERG at different ages. **j** Three-dimensional schematic showing lamina synapses (yellow box) formed by photoreceptors (green) and lamina neurons (L1–L2, red). Cross sections on the *xy*-plane and *xz*-plane are drawn for reference. **k** TEM micrographs of lamina cartridges (cross sections on the *xy*-plane; glia, blue; presynaptic terminals, green; post-synaptic terminals, red). **l** TEM micrographs of individual presynaptic terminals showing different types of membrane structures (type I-IV, indicated by asterisk) present in $dSms^{e/e}$ flies. **m** Quantification (mean ± S.D.; n = 32 for control and 29 for $dSms^{e/e}$ cartridges from three animals) of abnormal membrane structures observed in **k**. **n** TEM micrographs showing presynaptic mitochondria (green dashed outline) and post-synaptic mitochondria (red dashed outline) in lamina. **o** Quantification of presynaptic (green) and post-synaptic (red) mitochondria area (median with 10–90th percentile; n = 115/120 for control pre/post-synaptic terminals, and 93/79 for $dSms^{e/e}$ pre/post-synaptic terminals), number per cartridge (mean ± S.E.M., n = 25 for control and $dSms^{e/e}$ cartridges), and percentage of cartridges possessing mitochondria with abnormal morphology (mean ± S.E.M., n = 25 for control and $dSms^{e/e}$ cartridges). **g–i** One-way ANOVA post hoc Tukey test; **m, o** Student t test. *P < 0.05, **P < 0.01. All scale bars, 1 μm

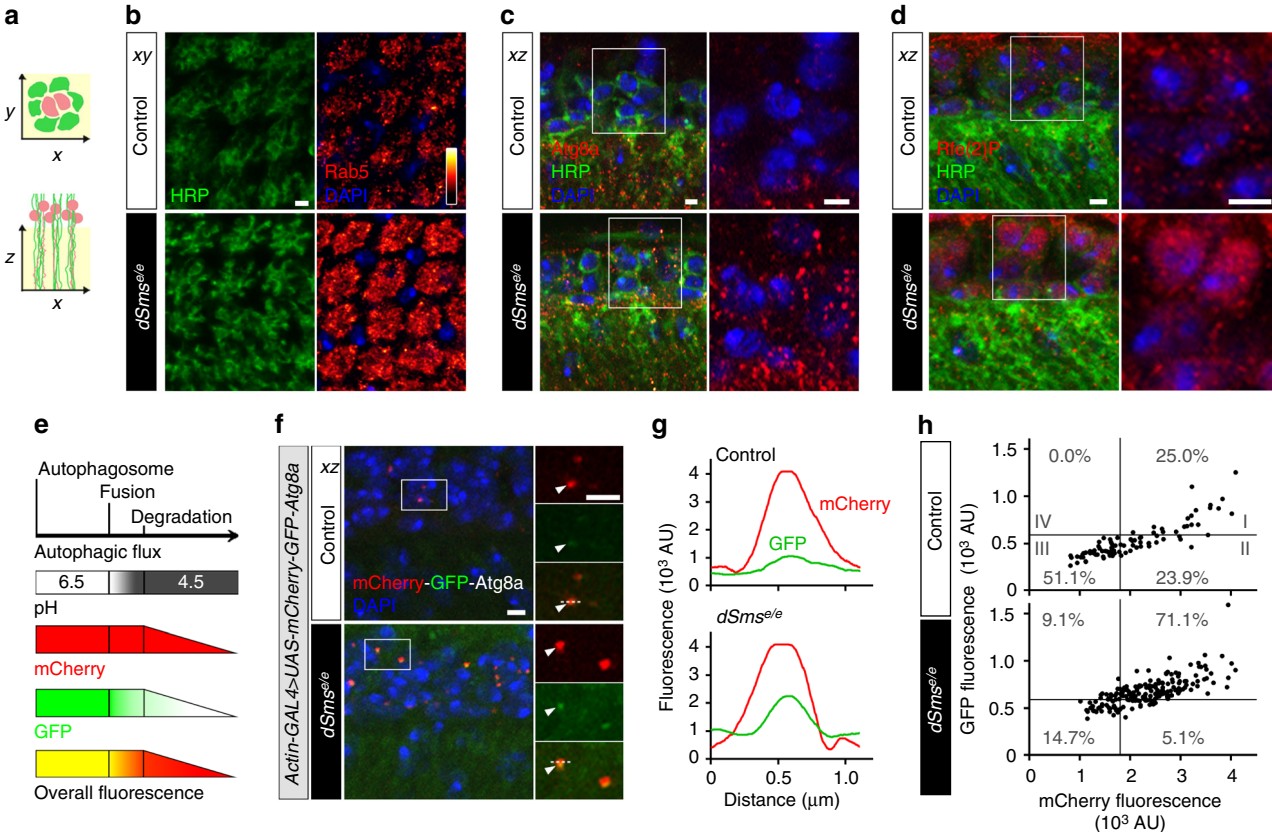

**Fig. 3** Loss of *dSms* impairs autophagy flux in *dSms* mutant synapses. **a** Cross sections of lamina synapses on the *xy*-plane and *xz*-plane are drawn for reference. **b** Confocal micrographs of immunostaining for early endosomes using marker Rab5 (intensity indicated with heat map). Neuronal membranes labeled with HRP and nuclei counterstained with DAPI. **c** Confocal micrographs of Atg8a immunostaining of endogenous phagophore/autophagosomes on 2 DAE fly laminae. Boxed area (left) indicates high magnification (right) of lamina neuronal cell bodies. **d** Confocal micrographs of Ref(2)P immunostaining of 2 DAE fly laminae. Boxed area (left) indicates high magnification (right) of lamina neuronal cell bodies. **e** Diagram showing the reporting mechanism of mCherry-GFP-Atg8a fusion protein. The pH of the autophagosome decreases from 6.5 to 4.5 upon fusion with the lysosome. GFP fluorescence is quenched in low pH environment, while mCherry fluorescence remains until the protein is degraded (slope). **f** Atg8a flux determined through overexpression of mCherry-GFP-Atg8a in control and *dSms^{e/e}* flies and visualized by confocal microscopy of lamina. Boxed area (left) indicates high magnification (right) of lamina neuronal cell bodies. Arrowheads indicate representative Atg8a-positive puncta plotted in **g** and quantified in **h**. **g** Histogram plot of the fluorescence intensity in arbitrary units (AU) along the dotted lines indicated in high-magnification images in **f**. **h** Scatter plot of Atg8-positive puncta measured from control (*n* = 4 animal) and *dSms^{e/e}* (*n* = 3 animal) animals (eight sections from each animal). The plot was divided into four quadrants using threshold of GFP = 570, mCherry = 1800 with the percentage of puncta in each quadrant indicated. All scale bars, 2 μm

restore lysosome integrity as measured by LAMP1 and cathepsin L immunolabeling (Supplementary Fig. 12). This is not surprising as lysosome defects are likely caused by aldehyde accumulation that is not targeted by these antioxidant compounds. Together, these results suggest that ROS accumulation contributes to the pathogenesis of SMS deficiency, which can be treated through boosting antioxidant defenses.

## Discussion

SRS is caused by mutations in *SMS* and loss of SMS enzyme activity. As pathogenic SMS proteins are defective in converting spermidine to spermine, reduced spermine levels are expected in SRS patients. Yet a lower spermine level is not consistently observed; instead, increased spermidine and a higher spermidine/spermine ratio are two features consistently observed in all SRS patients[14–18]. In fact, the spermidine/spermine ratio together with SMS enzyme activity have been used as the benchmark for SRS diagnosis since the genetic mapping of SRS to *SMS*[18]. Sustained spermine and increased spermidine in SRS patients underscore the complexity of polyamine metabolism and regulation and further implicate spermidine accumulation as the culprit of SRS

toxicity. Our studies in *Drosophila* dSms mutant nervous system and in SRS patient cells delineate the cellular consequences of pathological spermidine accumulation and illustrate mechanisms underlying SMS deficiency-induced neuronal dysfunction in SRS. Specifically, loss of SMS causes spermidine accumulation and catabolism, which generates aldehydes and the ROS $H_2O_2$. The accumulation of these cytotoxic metabolites impairs lysosome function and causes oxidative stress, which further compromises autophagy-lysosome flux, endocytosis, and mitochondrial function (Fig. 8).

Putrescine, spermidine, and spermine are all present in the mammalian brain, with different temporal and spatial distribution[50]. The function of polyamines in the nervous system has been inferred from in vitro observations, where polyamines were able to block or modulate several distinct types of cation channels, including inward-rectifier $K^+$ channels and glutamate receptors[51]. Such channel-modulating properties would necessitate a highly regulated local concentration of polyamines in the nervous system. Our data show that dSms is expressed in the nervous system, including cell bodies and synapses, and loss of dSms causes retinal and synaptic degeneration. These data suggest a critical requirement for local polyamine regulation

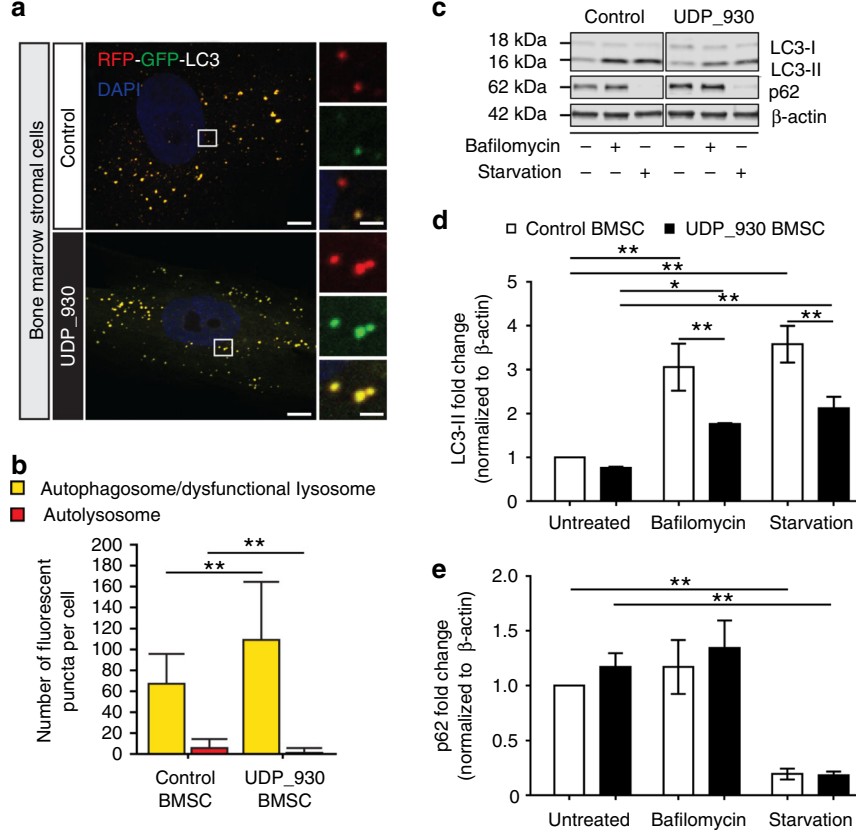

**Fig. 4** Autophagy flux is impaired in SRS BMSCs. **a** Confocal micrographs of LC3 flux examined through RFP-GFP-LC3 viral transduction in BMSCs. Boxed area (left) indicates magnified area (right) of autophagosomes and/or autolysosomes. **b** Quantification (mean ± S.D.; $n = 20$ cells per sample) of the numbers of autophagosomes/dysfunctional lysosomes (RFP- and GFP-positive puncta) and autolysosomes (only RFP-positive puncta). **c** Western analysis of cell extracts from BMSCs treated with or without autophagy inhibition (bafilomycin) or induction (starvation) with antibodies against LC3, p62, and β-actin. Full size blots with molecular weight markers are included in Supplementary Fig. 6. **d, e** Quantification (mean ± S.D.; $n = 3$ extractions) of LC3-II (**d**) and p62 (**e**) levels normalized to β-actin normalized to untreated control BMSCs. **d, e** One-way ANOVA post hoc Sidak (comparisons of preselected pairs) test; $*P < 0.05$, $**P < 0.01$

within neurons and support a compartmentalized impairment of neuronal functions caused by altered polyamine metabolism.

Our studies provide evidence of a direct link between spermidine catabolism and autophagic flux blockade. We showed that in dSms mutant flies, abnormal spermidine catabolism leads to increased $N^1$-acetyle spermidine ($N^1$-AcSpd) levels. This is consistent with spermidine accumulation in SRS and the recent report identifying acetylspermidine as a plasma biomarker for potential diagnosis of SRS[18,39]. Acetylated spermidine is oxidized by polyamine oxidase (PAO) or diamine oxidase, and both pathways produce aldehydes. Moreover, studies have shown that the amino-containing aldehydes generated during polyamine catabolism are lysosomotropic and cause lysosome dysfunction, partially through rupturing the lysosomal membrane and releasing lysosomal enzymes[32,43]. Clearly, the elevated aldehyde levels detected in dSms mutant flies, together with impaired lysosomal integrity and function observed in the synapses, are in agreement with aldehyde-induced toxicity. The accumulation of autophagic and endocytic vesicles in dSms mutant brain is consistent with the blockade of autophagy-lysosome flux due to downstream lysosome dysfunction. This detrimental consequence of elevated spermidine due to SMS deficiency contrasts the recent report demonstrating induction of autophagy by dietary spermidine through transcriptional and epigenetic regulation, which was shown to promote longevity and protect against age-induced memory loss[10,52]. It is likely that supplementing spermidine under physiological conditions with properly

regulated polyamine homeostasis is beneficial, whereas accumulation of spermidine in SMS deficiency leads to excessive polyamine catabolism that is pathological. Our studies highlight the importance of polyamine homeostasis and offer mechanistic insights into the cellular consequences of polyamine imbalance in disease conditions.

Our characterization of lysosomal defects sheds light on the pathogenesis not only of SRS, but also potentially for other neurological disorders with lysosomal involvement. Neurons are post-mitotic cells with extremely polarized morphologies. Thus, efficient clearance or recycling of diverse autophagic and endocytic substrates is required for neuronal function and maintenance[53], making the nervous system particularly vulnerable to lysosomal perturbation. Increasing attention is being paid to the relevance of lysosome dysfunction in neurodevelopment and neurodegenerative disease. This includes (1) complex diseases with prominent autophagic-endocytic-lysosomal neuropathology, such as Alzheimer's disease and Parkinson's disease[54,55] and (2) lysosomal storage disorders (LSDs), a heterogeneous group of monogenetic diseases with defects in lysosomal hydrolases or integral membrane proteins, such as Niemann-Pick disease type C and Pompe disease[56,57]. Intriguingly, symptoms common in early onset LSDs[56] including speech difficulties and seizures, as well as non-neurologic features including developmental delay and abnormal bone formation, are also prevalent in SRS patients[14], intimating a lysosomal origin of these phenotypes.

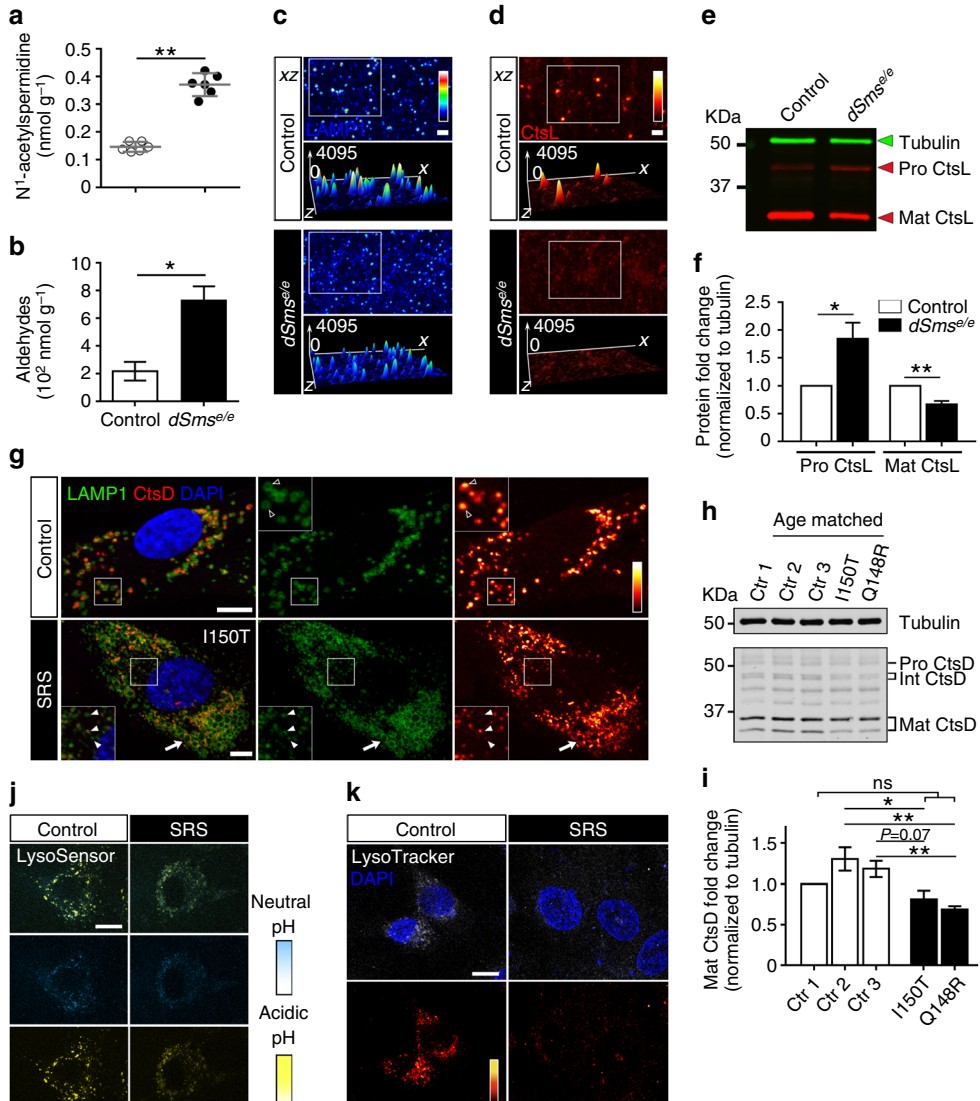

**Fig. 5** Spermidine oxidation impairs lysosome integrity and function in *dSms* mutant synapses and SRS fibroblasts. **a** N[1]-acetylspermidine level (mean ± 95% CI) measured from young flies (2–4 DAE) using LC-MS/MS. **b** Aldehyde content (mean ± S.E.M.; each data point obtained from one extraction of 20 animals, *n* = 3) measured from third instar larvae using a colorimetric method. **c**, **d** Immunostaining of fly laminae labeled with LAMP1 for lysosome membrane and cathepsin L for lysosomal protease. The fluorescence intensity profile is illustrated in a three-dimensional surface plot. LAMP1 (**c**) and cathepsin L (**d**) puncta staining intensity is compromised in mutant flies. **e**, **f** Western analysis of fly heads (**e**) and quantification (**f** mean ± S.E.M.; each data point obtained from one extraction of 10 animals, *n* = 7 extractions) showing pro- and mature forms of cathepsin L normalized to tubulin control. Full size blots with molecular weight markers are included in Supplementary Fig. 6. **g** Immunostaining of age-matched control and SRS fibroblasts labeled with LAMP1 and cathepsin D. Note both enlarged (arrow) and fragmented (white arrowhead) LAMP1 pattern presented in SRS fibroblasts. **h**, **i** Western blot analysis of human fibroblasts (**h**) and quantification (**i** mean ± S.E.M.; each data point obtained from one extraction, *n* = 5 extractions) showing pro-, intermediate, and mature forms of cathepsin D in SRS fibroblasts with two different mutations compared with control fibroblasts. Full size blots with molecular weight markers are included in Supplementary Fig. 6. **j**, **k** Live imaging of LysoSensor yellow/blue probe (**j**) and LysoTracker red probe (**k**) in human fibroblasts. Student's *t* test. *$P < 0.05$, **$P < 0.01$. Scale bar, **c**, **d**, 2 μm; **g**, **j**, **k**, 10 μm

Another byproduct of polyamine oxidation is hydrogen peroxide ($H_2O_2$), a ROS that leads to oxidative stress at high levels[58]. Oxidative damage and mitochondrial dysfunction are known to play critical roles in aging, neurodegeneration, and ischemic brain damage[59,60]. We also detected a remarkable elevation of ROS in dSms mutant brains. Oxidative damage can compromise mitochondrial integrity and if left unchecked can initiate a vicious cycle of deficient antioxidant balance leading to net ROS generation and further damage to mitochondria[60]. Until now, mitochondrial dysfunction has not been reported in SRS pathology. We observed morphological abnormalities of synaptic mitochondria, decreased ATP levels, and impaired COX activity

in SMS mutant *Drosophila* brain and muscle. In lymphoblastoid cells from SRS patients with 14 distinct SMS mutations, we observed a consistent shift toward glycolysis and away from mitochondria-dependent TCA cycle metabolism, suggesting mitochondria-related metabolic dysfunction. Our data suggest that polyamine oxidation causes oxidative stress and compromises mitochondrial function in the nervous system. Consistent with this finding, we observed beneficial effects from enhancing antioxidant capacity by genetic and pharmacological approaches. This observation is particularly important and encouraging for the SRS community since currently no pharmacological therapy is available.

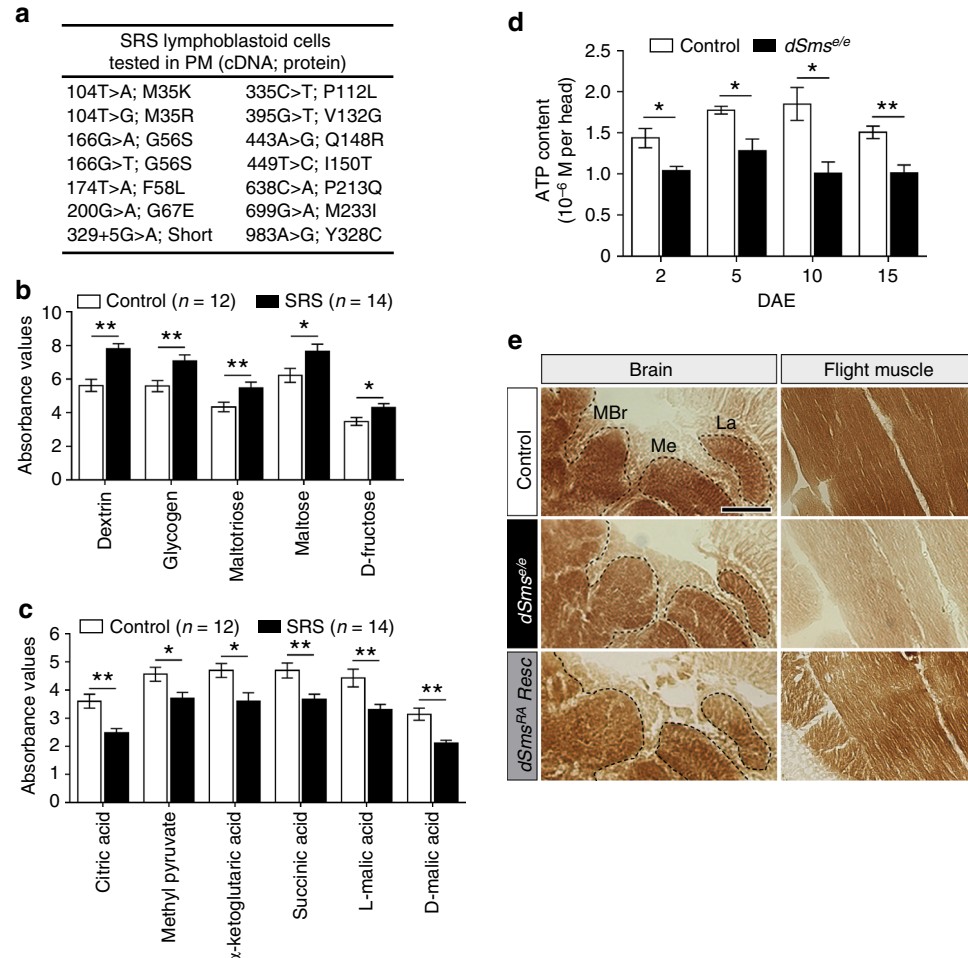

**Fig. 6** Mitochondria function is impaired in SRS cells and *dSms* mutant flies. **a** List of pathogenic SMS variants identified so far in SRS patients and tested in phenotype microarray. **b**, **c** Phenotype microarray profile (mean ± S.E.M.) showing the utilization of sugars (**b**) and compounds metabolized in the Krebs cycle (**c**) in lymphoblastoid cells from control ($n = 12$) and SRS patients ($n = 14$, indicated in **a**). **d** ATP content measured using bioluminescence assay from fly heads extracted at different ages (mean ± S.E.M.; $n = 5$ biological replicates, 10 heads each). **e** Histochemical analysis of COX activity in brain and flight muscle. Black dashed outline delineates different brain regions. MBr, middle brain; Me, medulla; La, lamina. Scale bar, 100 μm. Note decreased color intensity in the mutant group. **b**–**d** Student's *t* test; *$P < 0.05$, **$P < 0.01$

SRS is a multi-system disorder primarily affecting the nervous system and skeletal tissues[14–18]. SMS is widely expressed, however it is expected that different cells will have varied tolerance of SMS deficiency. Previous studies compared SRS patient cells, including bone-derived BMSCs, lymphoblastoid cells, and skin-derived fibroblasts, and found that although SMS mRNA and protein expression levels were similar, the cellular polyamine content was much more skewed in BMSC cells than in lymphoblasts or fibroblasts[14]. As lymphoblastoid cells and skin fibroblasts are less affected tissues compared to bone in SRS, the severity of polyamine imbalance is correlated with SRS pathogenesis[14–16]. To examine cell-type-specific mechanisms, we included three types of patient cells in this study, fibroblasts (Fig. 5 and Supplementary Figs. 5, 6, and 7), BMSCs (Fig. 4), and lymphoblastoid cells (Fig. 6a–c and Supplementary Fig. 8). Interestingly, our biochemical analysis of autophagy detected a significantly different extent of autophagic response assessed by LC3-II (Fig. 4c–e) in SRS patient-derived BMSCs compared to control BMSCs that were not detected in SRS fibroblasts (Supplementary Fig. 5). These results, together with our findings in *Drosophila* brain, support our model that polyamine imbalance and cellular defects explain the tissue-specific manifestation in SRS.

Here we report the cellular mechanisms underlying the toxicity induced by excessive spermidine catabolism in the nervous system. Our findings indicate that the polyamine imbalance reported to be associated with aging, neurodegeneration, brain injuries, and ischemia[4–8,32,43,61] is more than merely a result of these disease conditions, but could itself engender cellular damage. This is especially relevant when considering that spermine/spermidne-N[1]-acetyltransferase (SSAT), the rate-limiting enzyme in the polyamine oxidation pathway, is rapidly and highly induced, not only by polyamine alterations, but also a variety of stress or injury stimuli[37]. The induction of SSAT would putatively function to restore polyamine balance, but would likely be accompanied by a concomitant accumulation of metabolites from polyamine oxidation. Our characterization of spermidine catabolism-induced neurotoxicity provides insights for understanding the complexity of polyamine biology and will aid the design of antioxidant and aldehyde-neutralizing therapy for polyamine-associated neurological disorders.

## Methods

***Drosophila* stocks and genetics**. Unless specified, flies were maintained on a cornmeal-molasses-yeast medium at 25 °C, 65% humidity, 12 h light/12 h dark. The following fly strains were used in the studies: *actin-GAL4, elav-GAL4, UAS-mCherry-GFP-Atg8a, GstE1EP2231* obtained from Bloomington *Drosophila* Stock Center; *CG4300e00382* obtained from the Exelixis Collection at the Harvard Medical School; *dSms-GFPCB04249* obtained from Dr Spradling's lab (FlyTrap

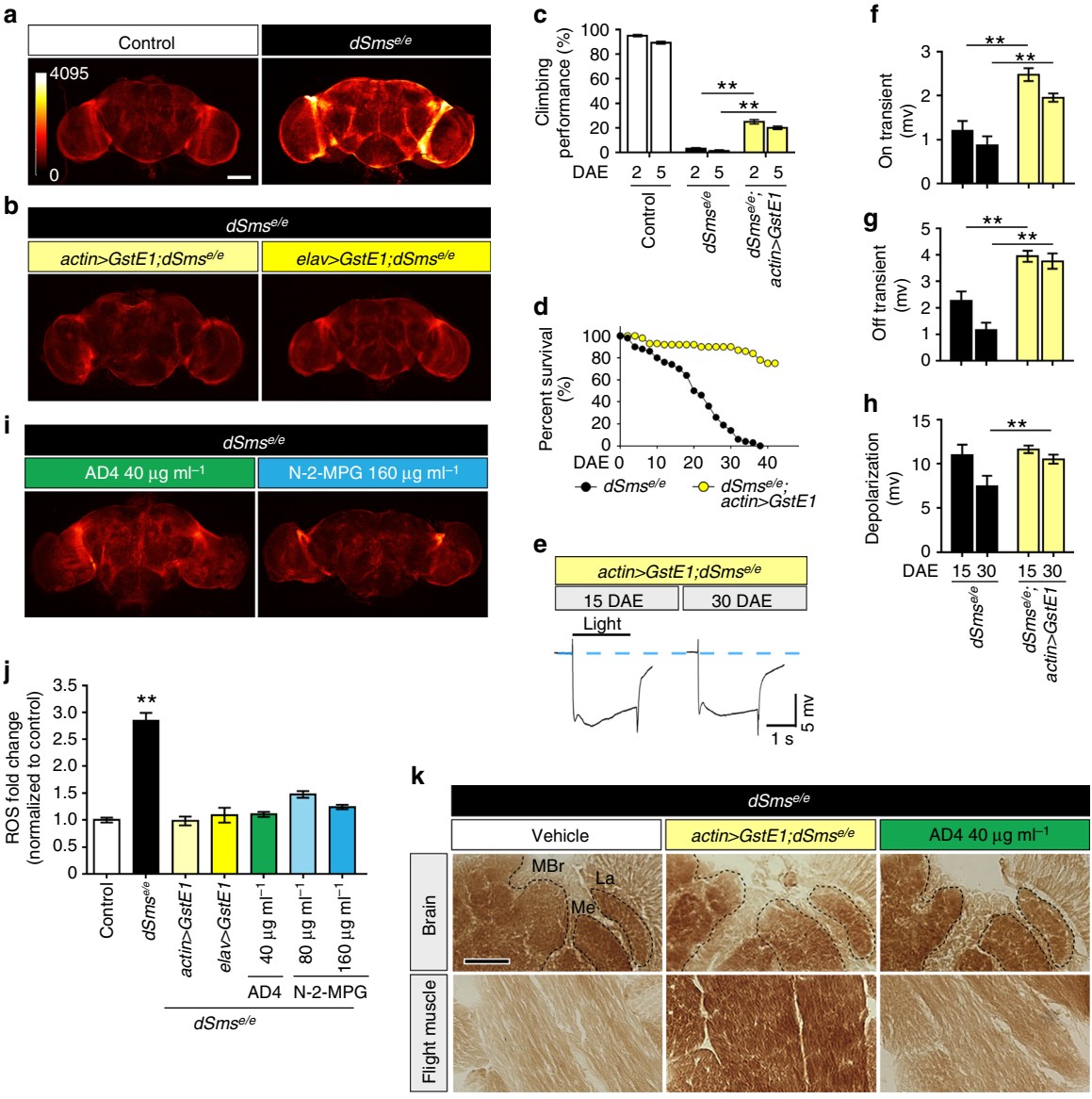

**Fig. 7** Oxidative stress caused by spermidine catabolism is suppressed through genetic or pharmacological enhancement of antioxidant capacity. **a**, **b** DHE staining (intensity indicated with heat map) showing ROS levels in fly brains. Note the increased fluorescence intensity in the mutant brains, which can be suppressed by ubiquitous (*actin-gal4*) or neuronal (*elav-gal4*) overexpression of GstE1. Quantified in **f**. Scale bar, 100 μm. **c** Climbing performance of control, $dSms^{e/e}$, and $dSms^{e/e}$ flies ubiquitously overexpressing *GstE1* at of 2 and 5 DAE (mean ± S.E.M.; each data point obtained from a group of 10 animals, $n = 10$ experiments). **d** Fly survival curves ($n = 62$ for $dSms^{e/e}$ and 66 for $GstE1$ rescue flies) determined by the age-specific number of live individuals. **e** Representative traces from ERG recordings showing light-induced depolarization and on/off responses. **f**–**h** Quantification (mean ± S.E.M.; $n = 10$ ($dSms^{e/e}$, 15 DAE), 10 ($dSms^{e/e}$, 30 DAE), 11 ($actin > GstE1$, 15 and 30 DAE) field recordings from 4–5 animals) of on transient (**f**), off transient (**g**), and depolarization (**h**) of ERG at different ages. **i** DHE staining showing ROS levels in brains of 5 DAE mutant flies raised on antioxidants at different concentrations. See **a** for staining of untreated mutant flies. **j** Quantification of ROS levels ($n = 12$ (control), 12 ($dSms^{e/e}$), 5 ($actin > GstE1$), 5 ($elav > GstE1$), 9 (AD 40 μg ml$^{-1}$), 5 (N-2-MPG 80 μg ml$^{-1}$), 5 (N-2-MPG 160 μg ml$^{-1}$) animals from each group; mean ± S.E.M.) normalized to control brain. **k** Histochemical analysis of COX activity in brain and flight muscle. Black dashed outline delineates different brain regions. MBr middle brain, Me medulla, La lamina. Scale bar, 100 μm. **c** Student's *t* test; **f**–**h**, **j** One-way ANOVA post hoc Tukey test. **\*\*$P < 0.01$

project[62]). The following plasmids were generated to make the transgenic flies: pTFW-hSMS$^{wt}$, pTFW-hSMS$^{443}$, pTFW-dSms$^{RA}$.

**Antibodies and reagents**. The following commercially available antibodies were used: anti-Rab5 (ab31261, abcam), anti-GABARAP for Drosophila Atg8a (PM037, MBL), anti-*Drosophila* LAMP1 (ab30687, abcam), anti-human LAMP1 (AB2296838, DSHB), anti-cathepsin D (sc-6487, Santa Cruz), anti-Brp (AB2314866, DSHB), anti-cathepsin L (MAB 22591, R&D systems)[44,63], anti-Ref (2)P[28], anti-elav (AB528217, DSHB), anti-ATP5α (ab14748, abcam), Cy5-conjugated anti-HRP (123175021, Jackson ImmunoLab), and secondary antibodies conjugated to Alexa 488/568/647 (Thermo Fisher Scientific), or near infrared (IR)

dye 700/800 (Rockland). The following chemicals were used in the study: N-acetylcysteine amide (A0737, Sigma-Aldrich), N-(2-mercaptopropionyl)glycine (M6635, Sigma-Aldrich), putrescine (51799, Sigma-Aldrich), spermidine (S0266, Sigma-Aldrich), spermine (s4264, Sigma-Aldrich), bis(hexamethylene)triamine (421960, Sigma-Aldrich), DHE (D11347, Thermo Fisher Scientific), LysoTracker® Red DND-99 (L7528, Thermo Fisher Scientific), LysoSensor™ Yellow/Blue DND-160 (L7545, Thermo Fisher Scientific).

**Fly viability and survival experiments**. Heterozygous $dSms^e$ flies recombined with *Actin-Gal4* or *UAS-SMS* transgenes were selected as parental flies for crossing (*Actin-Gal4; dSms$^e$*/TM3 x *UAS-SMS; dSms$^e$*/TM3). A total of 80–100 embryos

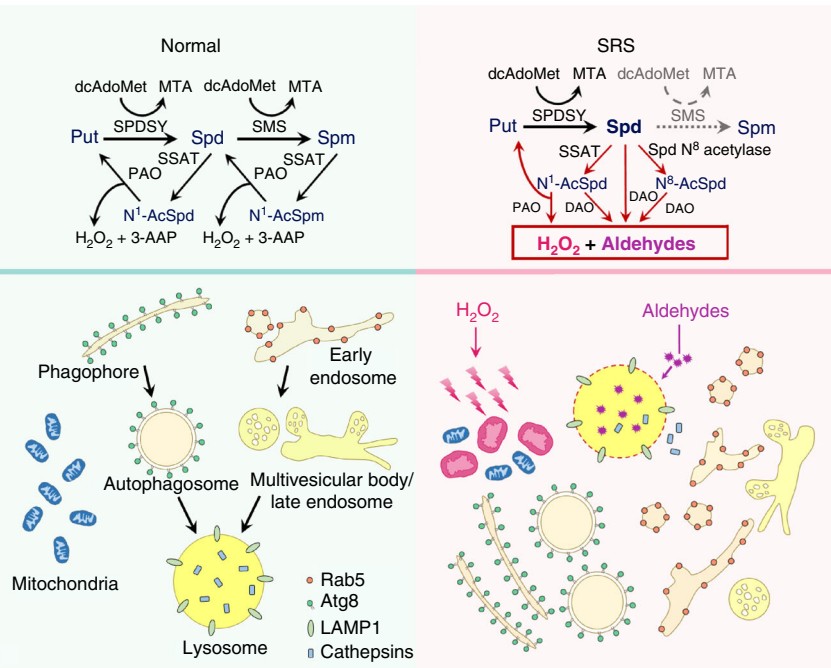

**Fig. 8** Diagram of cellular responses to spermidine catabolism caused by SMS deficiency in SRS. SMS is an aminopropyltransferase for catalyzing spermidine into spermine. SMS deficiency leads to spermidine buildup, followed by the induction of acetylation and oxidation. The accumulated polyamine catabolism metabolites, $H_2O_2$ and aldehydes, cause oxidative stress and lysosomal dysfunction, which further impair autophagy flux, endocytosis, and mitochondria function

were collected in each vial. Viability was calculated as the ratio of the homozygous to heterozygous $dSms^e$ flies expressing transgenes that survive to adulthood. For lifespan tracking, newly enclosed flies were collected and 20 flies of the same sex from each group were kept in fresh vials at 22 °C, 65% humidity. Flies were transferred to new vials every 3 days and the number of live flies was counted every other day.

**RNA extraction and quantitative PCR**. Total RNA was extracted from fly heads using TRIzol reagent (Invitrogen) according to the manufacturer's protocol. Each extraction was performed with 10 fly heads. RNA concentration was measured spectrophotometrically at 260 nm, and 2 μg of RNA from each sample was subjected to reverse transcription reaction with a high-capacity cDNA reverse transcription kit (Applied Biosystems). Samples were amplified with an amplification mix (20 μl) containing 100 ng of ssDNA reverse transcribed from total RNA and 1 μl of gene-specific TaqMan probe-primer set. The quantification was performed using 2(-Delta Delta C(T)) method and the mRNA levels were normalized to a housekeeping gene *rp49*.

**Polyamine extraction and measurements using LC/MS/MS**. Samples were collected from flash frozen flies stored at −80 °C. Approximately 12 mg of whole flies were semi-homogenized and extracted with 500 μl 5% trichloroacetic acid (TCA) and 100 μl internal standard (IS) using three repeated freeze-thaw cycles. For standard curves, 200 μl standard solutions were mixed with 300 μl 5% TCA and 100 μl IS. Samples were then centrifuged at 13,500×$g$ for 15 min. Ammonium acetate (0.4 M final concentration) was added to supernatants and samples were stored at −80 °C until polyamine determination using LC/MS/MS.

Polyamines were quantified by liquid chromatography tandem mass spectrometry (LC/MS/MS). The LC/MS/MS system consisted of a Shimadzu CBM-20A Controller, two LC-30AD pumps, SIL-30AC autosampler, CTO-30AC column oven, and an AB Sciex Triple Quad™ API 4500 mass spectrometer with turbo ion spray. Chromatographic separation was achieved with a Hypersil GOLD aQ column (4.6 × 250 mm, 5 μm, Thermo Scientific). The elution solvent A was 50 mM ammonium formate with 0.1% methanoic acid in ultra-pure water and elution solvent B was acetonitrile. Gradient separation was performed with 15% acetonitrile for 2 min. Then the concentration of acetonitrile was decreased to 5% within 2 min and increased to 15% within another 4 min, followed by column equilibration for 2 min. Flow rate was set at 500 μl min$^{-1}$ and column temperature was set at 25 °C. The following Q1/Q3 transitions were used: putrescine ($m/z$ 89.0 → 72.2, CE 12 eV), spermidine ($m/z$ 146.0 → 72.1, CE 20 eV), spermine ($m/z$ 203.3 → 129.2, CE 17 eV), N$^1$-acetylspermidine ($m/z$ 260.9→ 182.9, CE 14 eV), and bis(hexamethylene)-triamine as internal standard ($m/z$ 216.0 → 100.2, CE 25 eV).

**Electroretinogram analysis**. ERG analysis was performed as described[64,65]. Briefly, flies were anesthetized and glued on a glass slide, and the thorax, legs, and wings were immobilized. A recording electrode filled with 3 M KCl solution was placed on the compound eye and another reference electrode was inserted into the thorax. After 1–2 min adaption in the dark, flies were given a white light stimulus and the responses were recorded. The ERG traces were collected and analyzed using AxoScope 10.5 software.

**Retina sections and TEM**. Flies were dissected and fixed at 4 °C in freshly made solution containing 2% paraformaldehyde, 2% glutaraldehyde, 0.1 M sodium cacodylate, and post-fixed overnight in 2% OsO$_4$. After dehydration with graded ethanol solutions and propylene oxide (PO), specimens were infiltrated sequentially with mix of PO and resin. Specimens were then embedded in flat embedding molds and polymerized at 60 °C for 48 h. Thick sections (200–300 nm) of retina were cut on Leica Ultramicrotome with a glass knife and stained with toluidine blue stain. Thin sections (50 nm) were cut on Leica Ultramicrotome with a diamond knife and stained with 4% uranyl acetate in 100% methanol, and 0.25% lead citrate. JEOL JEM 1400 TEM and GATAN camera (Orius SC200 CCD) was used for ultrastructural examination.

**Lamina dissection and immunohistochemical staining**. Flies were dissected in cold PBS (pH = 7.4) and brains with attached lamina were fixed in freshly made 4% formaldehyde for 15 min. After 10 min washing in PBS containing 0.4% (v/v) Triton X-100 (PBTX) for three times, brains were incubated at 4 °C overnight with primary antibodies diluted in 0.4% PBTX containing 5% goat serum. Brains were then incubated at room temperature with conjugated secondary antibodies for 2 h, followed by staining with DAPI for 10 min. After washing, tissues were mounted on glass slides with VECTASHIELD Antifade Mounting Medium (Vector Laboratories) and kept at 4 °C until imaging.

**Fibroblast cell culture and immunocytochemistry**. Fibroblast cells were cultured in Chang Medium D (Irvine Scientific) supplemented with 1% antibiotic (Sigma-Aldrich) and fetal bovine serum (final concentration 20%, Atlanta Biologicals) at 37 °C with 5% CO$_2$.

Cells were seeded in 35-mm-diameter glass bottom culture dishes (MatTek Corporation) and allowed to grow for 24 h at 37 °C in growth medium. For immunocytochemical analysis of LAMP1 and cathepsin D, the cells were then fixed in freshly made 4% paraformaldehyde in warm PBS for 10 min, followed by 5 min post-fix in 100% methanol at −20 °C. After washing in PBS, cells were permeabilized and blocked with PBS containing 0.05% saponin and 5% bovine serum albumin (same solution for diluting antibodies) for 1 h at room temperature. Cells were then incubated with primary antibodies at 4 °C overnight and

conjugated secondary antibodies for 2 h at room temperature. After washing, cells were mounted with ProLong Gold antifade reagent with DAPI (Life Technologies) and allowed to cure for 24 h at room temperature in the dark before imaging.

**LysoSensor and Lysotracker live imaging**. For live imaging, LysoSensor or Lysotracker probes were added in pre-warmed growth medium and cells were incubated at 37 °C for 20 min. After washed twice with pre-warmed growth medium, the cells were immediately visualized using a confocal microscope.

**Confocal image acquisition and processing**. Specimens were imaged using an Olympus IX81 confocal microscope coupled with ×10, ×20 air lens or ×40, ×60, ×100 oil immersion objectives. Images were processed using FluoView 10-ASW software (Olympus) and analyzed using ImageJ software. Specifically, histogram plots in Fig. 3g and Supplementary Fig. 7e are analyzed using ImageJ Plot Profile and surface plots in Fig. 4c, d and Supplementary Fig. 12 are plotted using ImageJ Interactive 3D Surface Plot plugin.

**Immunoblot analysis**. For immunoblot analysis of cathepsins, tissues were homogenized on ice in lysis buffer containing 20 mM HEPES, 100 mM NaCl, 1 mM $CaCl_2$, 0.5% Triton X-100 and cOmplete protease inhibitor cocktail (Roche), followed by 10 min centrifugation at $10,000 \times g$ at 4 °C. The supernatants were then mixed with Laemmli sample buffer and heated at 95 °C for 10 min. Proteins were separated on a Bis-Tris 10% gel and transferred to a nitrocellulose membrane. After blocking, the membrane was incubated with primary antibodies overnight at 4 °C and near infrared dye-conjugated secondary antibodies for 2 h at room temperature. Imaging was carried out on an Odyssey Infrared Imaging system (LI-COR Biosciences) and images were analyzed using Image Studio software.

**Biochemical analysis of human fibroblasts and BMSCs**. To determine protein levels of the autophagic cargo p62 or the autophagosome marker LC3-II at baseline and upon inhibition or induction of autophagy, dermal fibroblasts were treated with the vehicle DMSO or treated with the autophagy inhibitor Bafilomycin A1 (400 nM, B1793, Sigma-Aldrich, St. Louis, MO) or the autophagy inducer rapamycin (1 μM, 37094, Sigma-Aldrich, St. Louis, MO) for 2 h, while BMSCs were untreated or treated with the autophagy inhibitor Bafilomycin A1 (100 nM, B1793, Sigma-Aldrich, St. Louis, MO) or starved with Earle's Balanced Salt Solution (24010043, Thermo Fisher Scientific, Waltham, MA) to induce autophagy for 2 h. Cells were trypsinized and resuspended in culture medium and centrifuged at 1000 rpm for 5 min. Cell pellets were washed with 1× PBS and lysed with RIPA buffer (R0278, Sigma-Aldrich, St. Louis, MO) supplemented with cOmplete, Mini, EDTA-free Protease Inhibitor Cocktail (Sigma-Aldrich, St. Louis, MO) on ice for 30 min. 4× Laemmli sample buffer (Bio-Rad Laboratories, Hercules, CA) and β-mercaptoethanol (M3148, Sigma-Aldrich, St. Louis, MO) were added and protein lysates were heated at 95 °C for 10 min. Proteins were resolved on a 4–15% Mini-PROTEAN TGX Stain-Free Gel (4568085, Bio-Rad Laboratories, Hercules, CA) and transferred to a PVDF membrane. After blocking in Odyssey Blocking Buffer (LI-COR Biosciences, Lincoln, NE) at room temperature for 1 h, the membrane was incubated with the primary antibodies anti-LC3B (L7543, Sigma-Aldrich, St. Louis, MO) and anti-alpha tubulin antibody (DM1A, ab7291, Abcam, Cambridge, MA), or anti-p62 (P0067, Sigma-Aldrich, St. Louis, MO) and anti-β-actin (AC-15, ab6276, Abcam, Cambridge, MA) diluted in Odyssey Blocking Buffer at 4 °C overnight. The membranes were then incubated with near infrared fluorescent dye-conjugated secondary antibodies at room temperature for 1 h. Imaging was performed on an Odyssey CLx Imaging System (LI-COR Biosciences, Lincoln, NE) and images were analyzed using the CLx Image Studio Version 3.1 Software (LI-COR Biosciences, Lincoln, NE).

**RFP-GFP-LC3B assay**. The Premo Autophagy Tandem Sensor RFP-GFP-LC3B Kit (P36239, Thermo Fisher Scientific, Waltham, MA) was used to assess autophagic flux in the patient dermal fibroblasts and BMSCs. Briefly, $1 \times 10^4$ cells were plated in each well of a 24-well plate and cultured overnight. Cells were then transduced with the BacMam 2.0 RFP-GFP-LC3B reagent at 40 particles per cell for 24 h. The autophagy inhibitor chloroquine (50 μM for 4 h, Thermo Fisher Scientific, Waltham, MA) was used as a positive control for the generation of autophagosomes and the autophagy inducer rapamycin (1 μM for 2 h, 37094, Sigma-Aldrich, St. Louis, MO) was used as a positive control for the generation of both autophagosomes and autolysosomes. After viral transduction, the cells were fixed in 4% paraformaldehyde for 15 min, stained with Hoechst 33342, and mounted with ProLong Gold Antifade Mountant (Thermo Fisher Scientific, Waltham, MA) and allowed to cure for 24 h prior to imaging. Cells were imaged using a Zeiss LSM 700 Confocal Microscope coupled with a C-Apochromat 40×/1.20 W Korr M27 objective. Images were processed using the ZEN software and analyzed using ImageJ.

**Aldehyde quantification assay**. Wandering third instar larvae were collected and washed with PBS. After quick freezing, samples were stored in −80 °C. For each measurement, 20 female larvae from each genotype were weighed. Larvae were homogenized in lysis buffer containing 0.5% (v/v) IGEPAL CA-630

(Sigma-Aldrich) on ice for 2 min, then centrifuged twice at low speed (3500 rpm) for 3 min at 4 °C to clear debris. Aldehyde determination was performed using an aldehyde quantification assay kit (ab112113, Abcam) that uses a proprietary dye that generates a chromogenic product upon reacting with an aldehyde. Absorbance spectroscopy was measured with a FLUOstar Omega microplate reader (BMG Labtech). The optical density at 550 nm was used for quantification and the results were normalized to body weight. Reaction mix without sample was used as a blank control.

**ATP bioluminescence assay**. ATP content was measured using ATP biolumi-nescence HSII assay kit (Roche) as described[66]. Briefly, fly heads were dissected and homogenized using lysis buffer (40 μl per head) on ice for 1 min. Samples were then boiled for 5 min at 100 °C, followed by centrifugation for 2 min at 14,000 rpm. Supernatants were transferred to new tubes and centrifuged for another 1 min. Supernatants were transferred again for additional centrifugation for 2 min at 14,000 rpm at 4 °C. Supernatants were then diluted with dilution buffer and 25 μl was placed in triplicate in a 96-well plate. Luciferase activity was detected by a FLUOstar Omega microplate reader (BMG Labtech) with an automated injection system, allowing for simultaneous reagent injection and detection. An aliquot of 25 μl of luciferase reagent was injected and luminescence signal was monitored over a period of 12 s (kinetic method).

**COX histochemistry and quantification**. COX activity was measured as descri-bed[67,68]. Briefly, 10-μm fresh ultra-rapid frozen head and flight muscle sections were collected and stored at −20 °C until ready to use. Samples were dried at room temperature for 15 min, and 100 μl of staining solution (0.1% 3′-diaminobenzidine, 0.1% cytochrome c, 0.02% catalase in 5 mM PBS) was added to each section. Samples were incubated for 20 min at room temperature and washed with PBS (pH = 7.4) to stop reaction. The samples were then dehydrated for 2 min in each of the following concentrations of ethanol: 70, 70, 95, 95, and 99.5%, followed by additional 10 min dehydration in 99.5% ethanol. Slides were then placed in xylene for 10 min. The slides were covered with a coverslip and allowed to dry overnight before imaging. Quantification was carried out using ImageJ software. For each section, four–five circle areas along the flight muscle were selected as ROIs. The area size and raw integrated density from each ROI were captured and COX activity is calculated by area size/raw integrated density. For each section, four–five ROIs were averaged to give one data point.

**ROS detection and quantification**. ROS detection was performed as described[69]. Briefly, 1 mg of DHE was dissolved in 100 μl DMSO and then diluted in Schneider's medium (SM) right before use. Flies were dissected in SM at room temperature and the brains were incubated with 30 μM DHE in the dark for 15 min. After washing four times with PBS, brains were immediately mounted on glass slides with VECTASHIELD Antifade Mounting Medium (Vector Labora-tories). Imaging was performed within 4 h. Quantification was carried out using ImageJ software. Brains were scanned top to bottom using the same fluorescence intensity and step size. Max z stack projections were used to set a threshold to select an ROI including the entire brain area. A minimum threshold was applied to all images and anything below this limit was considered to be background noise. For each brain, threshold values in arbitrary units of the ROI from all sections were added to give one data point.

**Drug administration**. Jazz-Mix™ *Drosophila* food (Fisher Scientific) was dissolved in purified water and boiled for 10 min. When the temperature cooled to 50 °C, the food was thoroughly mixed with drug solutions, transferred into new vials, and allowed to solidify for 30 min. Eggs were laid on fresh food supplemented with drug or solvent and raised at 25 °C. Newly enclosed adults were transferred to new corresponding vials until experiments were complete.

**Negative geotaxis behavior assay**. Ten age-matched male or female flies from each genotype were placed in a vial marked with a line 8 cm from the bottom surface. The flies were gently tapped onto the bottom and given 10 s to climb. After 10 s, the number of flies that successfully climbed above the 8 cm mark was recorded and divided by the total number of flies. The assay was repeated 10 times, and 10 independent groups (total 100 flies) from each genotype were tested.

**Biolog metabolic arrays**. Biolog metabolic arrays were performed as previously described[70]. Each well of the PM-M plates contains a single chemical as the only energy source, and using colorimetric redox dye chemistry, the generation of NADH in each well is monitored. The plates were incubated with 20,000 lymphoblastoid cells per well in a volume of 50 μl. The cells were incubated for 48 h at 37 °C in 5% $CO_2$, using the modified Biolog IF-M1 medium, to 100 ml, 1.1 ml 100 × penicillin/streptomycin solution, 0.16 ml 200 mM glutamine (final concentration 0.3 mM), and 5.3 ml fetal bovine serum (final concentration 5%). Following the first incubation, Biolog Redox Dye Mix MB was added (10 μl per well), and the plates were incubated for a further 24 h. During this time, the cells metabolize the sole carbon source in the well. After the 24 h incubation, the plates were analyzed utilizing a microplate reader with readings at 590 and 750 nm.

The first value (A590) indicated the highest absorbance peak of the redox dye and the second value (A750) gave a measure of the background noise. Relative absorbance (A590-750) was calculated per well. For the kinetic curves, Biolog plates were placed in an Omnolog apparatus for 24 h after addition of the redox dye. Quantitative color change values were captured every 15 min and displayed in a form of kinetic graphs.

**Bioinformatics**. Multiple sequence alignment was performed using ClustalW. The following sequences are used: *Homo sapiens* (NP_004586.2), *Mus musculus* (NP_033240.3), *Gallus gallus* (NP_001025974.1), *Danio rerio* (NP_571831.1), *Drosophila melanogaster* (NP_729798.1), *Saccharomyces cerevisiae* (NP_013247.1). RNA-Seq expression levels were obtained from FlyBase GBrowse.

**Statistics**. No statistical methods were used to predetermine sample size. Investigators were blinded for TEM and ROS data collection and quantification. Data were analyzed with Prism (GraphPad Software). Student's $t$ test (two tailed) was used for comparison of two groups. One-way ANOVA with Bonferroni ($n < 5$), Tukey ($n \geq 5$), or Sidak (comparisons of preselected pairs) correction was used for comparison of more than two groups. $P < 0.05$ was considered statistically significant.

**Data availability**. The data that support the findings of this study are available within the article and Supplementary Files or available from the corresponding author on request.

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

## Acknowledgements

We are grateful to all SRS families. We thank T. Koerner, M. Raymond, and all members of the Snyder-Robinson foundation for their support. We thank Y. Moon, J.E. Balke, Y. Meng, Y. Huang (NIH-UDP), and J. Guo (NIH-UDP) for technical assistance; A. Barrientos for reagents; G. Juhasz and E.H. Baehrecke for Reference(2) P antibody, M.R. Bates and V.W. Almeida from the Electron Microscopy Core Facility at the University of Miami for their assistance; K. Ruan and J.S. Park for technical suggestions and discussions. This work is supported by the Snyder-Robinson Foundation Predoctoral Fellowship (to C.L.), the Dr. John T. Macdonald Foundation (to C.L.), the Lois Pope LIFE Fellows Program (to C.L., Y.Z., and J.M.B.), the Sheila and David Fuente Neuropathic Pain Research Program Graduate Fellowship (to J.M.B.), contracts, grants from National Institutes of Health (NIH) HHSN268201300038C, HHSN268201400033C, and R21GM119018 (to R.G.Z.), and by Taishan Scholar Project (Shandong Province, People's Republic of China).

## Author contributions

C.L. and R.G.Z. designed the experiments and wrote the manuscript with input from J.M.B. and all authors. W.A.G., C.F.B., and R.G.Z. conceived the experimental approach. C.L. carried out all experiments in *Drosophila* with help from J.M.B., C.B., Y.Z. and Z.D.P. S.L. and H.W. carried out LC/MS analyses. For the phenotype microarray analyses, L.C. and L.B. performed the assays, R.P. carried out the statistical analyses, and C.E.S. conceived the experimental approach and interpreted the data. M.M. and M.C.V.M. carried out autophagy analysis in SRS patient cells. J.M.B., C.E.S, C.F.B., W.A.G., and R.G.Z. edited the manuscript.

## Additional information

**Competing interests:** The authors declare no competing financial interests.

**Change history:** A correction to this article has been published and is linked from the HTML version of this paper.

