## [Peer Review File · Nature Communications]

Reviewers' comments:

Reviewer #1 (Remarks to the Author):

In this paper, Li et al. report on the phenotype of flies lacking spermine synthase, an enzyme that is mutated in human Snyder-Robinson X-linked Intellectual Disability Syndrome (SRS). The authors report that toxic polyamine metabolites cause autophagy-lysosome flux and mitochondrial function. The authors claim in the abstract that antioxidants reverse the phenotype in the fly model, yet fail to provide evidence in favor of this statement apart from the fact that transgenic expression of GstE1 improves viability and locomotor activity.

Altogether, the paper should be improved in the following points:

- It would be interesting to know whether external supply of spermidine might improve the phenotype of dSmse/e flies, knowing that such flies show a deficit in endogenous spermidine. Spermidine supply can improve the locomotor deficit of old flies and extend their lifespan. dSmse/e flies manifest a locomotor deficit and a reduced lifespan. So, it would be plausible that spermidine deficiency accounts for (part of) the phenotype.
- Phrases like "These results suggest that SRS dermal fibroblasts have increased autophagy secondary to impaired downstream autophagic flux." are extremely confusing. The authors should talk about the number of autophagosomes per cell rather than "autophagy". Similarly, the wording of "spermidine oxidation" may be incorrect, and it may be better to talk about "spermidine catabolism", the first step of which is indeed the acetylation of spermidine. But the acetylation reaction is not an oxidation reaction.
- SRS patient fibroblasts and dSmse/e flies apparently manifest a "compromised lysosomal integrity and function". It would be important to measure this by assessing lysosomal pH with lysosensors. It may also be interesting to use supplementat technologies to assess lysosomal membrane permeabilization (see Aits et al. Methods Cell Biol. 2015;126:261-85).
- The data in Fig. 6f show a tautology: antioxidants reduce oxidative stress. What is not shown, however, is whether genetic manipulations and pharmacological administration of antioxidants reverse the lysosomal, mitochondrial, behavioral, and longevity phenotype of dSmse/e flies. This is the crux of the paper.

Reviewer #2 (Remarks to the Author):

Li et al characterize a Drosophila model of SRS, an X-linked intellectual disability syndrome. Their findings indicate an unbalance of polyamine levels and increased spermidine, as expected by the mutation of spermine synthase in these cases. Surprisingly, an impairment of autophagy is also suggested, even though previous publications showed that dietary spermidine extends longevity via induction of autophagy. I think the autophagy analyses should be strengthened to support these claims.

Major comments

1. Fig 2c-e: are the sms mutant flies on a white+ background, or is the pale yellow coloration of

the eyes due to the transposon insertion in white- flies? If it is the latter case, then this experiment should be repeated in mutant animals with otherwise wild-type eye color, similar to controls shown in Fig 2a-b.

2. Cyto-ID-autophagy marker is seldom used for autophagy analysis, so its specificity is rather questionable (and I do not see that mutant cells take up less dye than controls upon chloroquine treatment in fig 3g, middle panels). I suggest that the authors repeat the tests shown in Fig 3g using Cherry-GFP-LC3B, the commonly used and specific marker of autophagosomes and autolysosomes. This would be important also because bafilomycin and rapamycin have no effect on LC3 lipidation in control cells (fig 3h).

3. A related comment is that the most common test for analyzing autophagic flux is by looking at the level of the specific autophagic cargo p62/SQSTM1 (Ref2P in flies). The level of endogenous p62 should be determined in control and mutant *Drosophila* and human cells, which will ideally support the suggested impairment of autophagic flux.

4. Fig 4d: there is a huge difference in proenzyme levels between control samples, and one of the lots of bands is marked as mature CtsD. This blot is not convincing: more controls and a proof of antibody specificity are needed, or it should be left out.

Minor comments

5. Transposon insert seems to be in an intron in Fig 1b, unlike in the end of exon 3 as stated in the text.

6. Typo in Fig 1g: putrescien

7. Source and identity of *Drosophila* anti-CtsL antibody is missing from the Methods section.

Reviewer #3 (Remarks to the Author):

The manuscript by Li et al. describes a *Drosophila* model for the Snyder Robinson X-LID syndrome. In their experiments, they identify the generation of H₂O₂ and amine-containing aldehydes as key mechanistic features of the disease resulting in lysosomal dysfunction, in-line with previous data, and also mitochondrial dysfunction which represents a novel aspect of the disease. The manuscript offers a significant advance in the understanding of Snyder Robinson X-LID syndrome and key findings are supported by analysis of patient cells. The experiments are largely (with some exceptions, see below) convincing and well done. A few aspects need to be addressed, however, before the manuscript is appropriate for publication.

- Several of the key experiments assessing *Drosophila* phenotypes are shown without rescue controls, despite those flies being available. While this may be reasonable for measurements of metabolites directly related to SMS activity (like spermidine and putrescin (in Figure 1) and their metabolites (in Figure 4A,B), it is not acceptable for the measurement of complex phenotypes (like longevity, negative geotaxis and neurodegeneration) which are easily influenced by different genetic backgrounds and require rescue experiments to unequivocally assign phenotypes to specific genes. As the necessary flies are available (See figure 1e) and the assays established, this should not be a significant problem delaying the manuscript.

- The claimed "enrichment" of SMS at synapses is not convincingly demonstrated. First, there are no data provided which show that the GFP trap line encodes a functional fusion protein, whose localization reflects that of the endogenous protein. Second, the low-resolution images provided

are just as consistent with distribution throughout cells as they are with enrichment at the Brp-labeled synapses. This claim needs to be better supported or could be removed without reducing the impact of the manuscript.

- The effects of Bafilomycin and Rapamycin on the LC3-I/-II ratios in Figure 3 h are not obvious and need to be quantified.

- The authors demonstrate a remarkable reduction of pathogenic ROS production by supplementing food with AD4 or N-2-MGP. Very disappointingly they do not show the phenotypic consequences on the tissue distribution of Cox activity (as shown in Fig. 5e) neurodegeneration or behavioral outcomes. Even if the outcomes are "negative", these would be important additions significantly strengthening the manuscript.

We would like to thank the reviewers for their excellent comments and suggestions. We have followed the recommendations and carried out several lines of experiments to address all the comments and concerns, and further substantiate the significance of our findings. Due to the extensive timeline of the genetic rescue and aging experiments, it took us more than 3.5 months to complete the revision of the manuscript.

As the revision is both comprehensive and extensive, here we include two tables –a list of new data figures included in the revision (Table 1) and a summary of main points raised by reviewers (Table 2) as an overview for the detailed point-by-point response to all of the issues raised. Major revisions in the results, methods and figure legends sections are marked by a blue font to highlight the new data. The revisions made throughout the rest of the manuscript are unmarked.

Table 1: Summary list of major changes to the figures.

	Changes/ Additions	Experiments	Responding to Reviewer
Figure 1	1h-i	Genetic rescue	Reviewer 3
Figure 2	2a-f, 2a'-f', and 2g-i	Neuronal phenotypes of rescue	Reviewer 3
Figure 3	3d	Autophagy (in vivo , Drosophila)	Reviewers 2
Figure 4	New	Autophagy (human patient cells)	Reviewers 2 and 3
Figure 5	5h-k	Lysosome phenotypes	Reviewers 1, 2 and 3
Figure 6	6e	Genetic rescue	Reviewer 3
Figure 7	7d-h, and 7k	Genetic and pharmacological rescue	Reviewers 1 and 3
Figure 8	Revised	Working model	Reviewer 1
Supplemental Figure 5	New	Autophagy (human patient cells)	Reviewers 2 and 3
Supplemental Figure 8	New	Genetic and pharmacological rescue	Reviewer 1 and 3
Supplemental Figure 9	New	Neuronal phenotypes of rescue	Reviewer 1
Supplemental Figure 10	New	Genetic and pharmacological rescue	Reviewer 1 and 3
Supplemental Figure 11	New	Lysosome phenotypes	Reviewers 1 and 3

Table 2: Summary of results from revision experiments carried out in response to key comments raised by all reviewers.

Key Comments	Reviewers			Remedy
	1	2	3	
1. Strengthen the lysosome function analysis	•			Examined the pH and integrity of lysosomes in SRS patient fibroblasts using live-cell dye labeling with LysoSensor (Fig. 5j) and LysoTracker (Fig. 5k)
2. Functional rescue of genetic manipulations	•			Examined the effects of GstE1 overexpression on dSms mutant phenotypes 1) Life span (Fig. 7e) 2) COX activity in fly brain and flight muscles (Fig. 7k, Sup. Fig. 10) 3) Function and morphology of the visual system (Fig. 7e-h, Sup. Fig. 9)
3. Functional rescue of pharmacological administration of antioxidants	•		•	Examined the functional effects of drug feeding on dSms mutant phenotypes 1) Viability (Sup. Fig. 8) 2) COX activity in brain and flight muscles (Fig. 7k, Sup. Fig. 10) 3) Lysosome integrity measured by LAMP1 and Cath L staining (Sup. Fig. 11)
4. Assess fly phenotypes with genetic rescue control			•	Examined the effects of dSms ^{RA} overexpression on dSms mutant phenotypes 1) Life span (Fig. 1i) 2) Climbing performance (Fig. 1h) 3) Function and morphology of the visual system (Fig. 2f-f') 4) COX activity in brain and flight muscles (Fig. 6e)
5. Strengthen the autophagy analysis		•	•	In Drosophila : 1) Performed brain Ref(2)P immunostaining (Fig. 3d) In SRS patient cells: added bone marrow stromal cells, performed all the following analysis on two cell types: 2) RFP-GFP-LC3 analysis (Fig. 4a,b, Sup. Fig. 5a,b) 3) LC3 biochemical analysis (Fig. 4c,d, Sup. Fig. 5c,d) 4) p62 biochemical analysis (Fig. 4c,e, Sup. Fig. 5e,f)
6. Claim of synaptic enrichment of dSms			•	Deleted the description in the main text. Provided a figure for the reviewer.
7. Eye color control of dSms mutants		•		Replaced CS (wild type) flies with UAS-dSms ^{RA} transgenic flies (parental control) 1) Eye morphology (Fig. 2a,b) 2) ERG (Fig. 2a',b')
8. CtsD western analysis on patient cells		•		Repeated with additional age-matched controls (Fig. 5h), added quantification (Fig. 5i)
9. Misuse of spermidine oxidation	•			1) Replaced with “spermidine catabolism” as suggested 2) Modified the model to further distinguish two toxic metabolites (Fig. 8)
10. ‘Spermidine’/spermine feeding	•			Examined the effects of spermine feeding on viability, brain ROS levels and lysosome integrity. Provided a figure for the reviewer.

Point-to-point response to reviewers' comments:

Reviewer #1 (Remarks to the Author):

In this paper, Li et al. report on the phenotype of flies lacking spermine synthase, an enzyme that is mutated in human Snyder-Robinson X-linked Intellectual Disability Syndrome (SRS). The authors report that toxic polyamine metabolites cause autophagy-lysosome flux and mitochondrial function. The authors claim in the abstract that antioxidants reverse the phenotype in the fly model, yet fail to provide evidence in favor of this statement apart from the fact that transgenic expression of GstE1 improves viability and locomotor activity.

We thank the reviewer for recognizing the significance of our findings. In this revised version, we have comprehensively characterized the cellular, biochemical, as well as functional effects of antioxidants on fly neurodegeneration (see responses to comment #4). We identified significant functional protection of antioxidant treatment by both genetic (GstE1 overexpression) and pharmacological (drug feeding) approaches.

Altogether, the paper should be improved in the following points:

1) It would be interesting to know whether external supply of spermidine might improve the phenotype of dSmse/e flies, knowing that such flies show a deficit in endogenous spermidine. Spermidine supply can improve the locomotor deficit of old flies and extend their lifespan. dSmse/e flies manifest a locomotor deficit and a reduced lifespan. So, it would be plausible that spermidine deficiency accounts for (part of) the phenotype.

We would like to first clarify that SRS is caused by loss of spermine synthase, an enzyme that converts spermidine into spermine. Therefore in dSMS mutant tissue, there is a buildup, rather than deficiency of spermidine (Fig. 1g). We thus thought the reviewer meant “a deficit in endogenous spermine”. It is important to note that in SRS patients, spermine deficiency was not consistently observed (see the first paragraph of the discussion), which propelled us to seek the underlying toxicity. Nevertheless, it will be interesting to examine the effects of supplementation with spermine. Thus, we performed spermine feeding experiments and examined the viability, brain ROS accumulation, and lysosome function, and included the results here for the reviewer.

[Redacted]

[Redacted]

Regarding spermidine, it has been shown that in wild type flies spermidine levels decline with aging, and spermidine supplementation can extend fly life span (Eisenberg et al., 2009) and restore aspects of learning and memory decline (Gupta et al., 2013). However, there are conflicting reports as to whether spermidine supplementation can rescue age-dependent decreases in locomotor functions (Gupta et al., 2013; Minois et al., 2014). Since loss of SMS causes spermidine buildup, we did not attempt spermidine supplementation as it is expected to further exacerbate the phenotypes.

2) Phrases like "These results suggest that SRS dermal fibroblasts have increased autophagy secondary to impaired downstream autophagic flux." are extremely confusing. The authors should talk about the number of autophagosomes per cell rather than "autophagy". Similarly, the wording of "spermidine oxidation" may be incorrect, and it may be better to talk about "spermidine catabolism", the first step of which is indeed the acetylation of spermidine. But the acetylation reaction is not an oxidation reaction.

We thank the reviewer for pointing this out. In this revised manuscript, we have revised the description of autophagic flux. Specifically, we have quantified the numbers of autophagosomes and the autolysosomes to reflect the defects in autophagic flux.

We agree with the reviewer that the use of spermidine oxidation is misleading. Specifically, the two toxic metabolites generated through spermidine catabolism have different cellular targets: H_2O_2 causes oxidative stress and aldehyde impairs lysosome function. We thus replaced the phrase 'spermidine oxidation' with 'spermidine catabolism' throughout the manuscript. In addition, we revised our model diagram to emphasize the distinction between two metabolites (**Fig. 8**). This word change has made our description more accurate and precise. We are grateful for this comment.

3) SRS patient fibroblasts and dSmse/e flies apparently manifest a "compromised lysosomal integrity and function". It would be important to measure this by assessing lysosomal pH with lysosensors. It may also be interesting to use supplementat technologies to assess lysosomal membrane permeabilization (see Aits et al. Methods Cell Biol. 2015;126:261-85).

To address this comment we have performed live imaging assays using LysoSensor™ Yellow/Blue probe on fibroblasts derived from SRS patients and age-matched controls. This LysoSensor dye exhibits dual-emission spectral peaks (blue fluorescence in neutral pH and yellow fluorescence in acidic pH) and the fluorescence intensity is pH-dependent (Pavel et al., 2016). We saw decreased yellow fluorescence in SRS fibroblasts, which clearly suggests a less acidic lysosome (**Fig. 5j**). In addition, we have applied LysoTracker® Red probe to further verify the integrity of the lysosomes (Merkulova et al., 2015). LysoTracker is a fluorescent dye that can freely permeate across cell membranes at neutral pH, but becomes protonated and trapped in acidic compartments. In control fibroblasts, we observed readily labeled acidic organelles. However, SRS fibroblasts showed diffuse LysoTracker dye labelling (**Fig. 5k**).

These results, together with our immunofluorescent staining using LAMP1 and cathepsin antibodies (**Fig. 5c, d, g**), as well as biochemical analysis of cathepsins (**Fig. 5e, f, h, i**), strongly support

compromised lysosomal integrity and function in SRS fibroblasts.

4) The data in Fig. 6f show a tautology: antioxidants reduce oxidative stress. What is not shown, however, is whether genetic manipulations and pharmacological administration of antioxidants reverse the lysosomal, mitochondrial, behavioral, and longevity phenotype of *dSmse/e* flies. This is the crux of the paper.

To address this point and a similar point raised by Reviewer 3 (see below), we have carried out additional experiments to comprehensively examine the effects of antioxidants by genetic manipulation and pharmacological administration on *dSms* mutant phenotypes.

Genetic manipulation: We overexpressed GstE1 in homozygous *dSms* mutant flies and found that 1) GstE1 overexpression significantly extended life span of *dSms* mutant flies (**Fig. 7d**); 2) GstE1 overexpression significantly rescued the pigmentation changes at DAE15 and DAE30 (**Supplementary Fig. 9**) and the retina physiology measured by ERG at DAE15 and DAE30 (**Fig. 7e-h**); and 3) GstE1 overexpression restored COX activity at DAE5, especially in fly flight muscles (**Fig. 7k, Supplementary Fig. 10**).

Pharmacological administration: We supplemented the food with antioxidant compounds at different concentrations and found that, in addition to reducing brain ROS accumulation, AD4 feeding increased COX activity suggesting partial restoration of mitochondria function (**Fig. 7k, Supplementary Fig. 10**). However, administration of AD4 (40 µg/ml) or N-2-MPG (160 µg/ml), did not restore lysosome integrity as measured by LAMP1 and cathepsin L immune-labeling (**Supplementary Fig. 11**). This is not surprising as lysosome defects are likely caused by aldehyde accumulation that is not targeted by these anti-oxidant compounds. We also examined the effects of antioxidant feeding on survival. *dSms* mutant flies have a reduced survival rate, that is less than 50% of the flies complete metamorphosis and eclosion process. Feeding *dSms* mutant larvae with antioxidant AD4 (40 µg/ml) did not significantly improve viability (**Sup. Fig. 8**), suggesting that feeding for three days at larval stage is insufficient to overcome metamorphosis defects and improve eclosion rate.

Reviewer #2 (Remarks to the Author):

Li et al characterize a *Drosophila* model of SRS, an X-linked intellectual disability syndrome. Their findings indicate an unbalance of polyamine levels and increased spermidine, as expected by the mutation of spermine synthase in these cases. Surprisingly, an impairment of autophagy is also suggested, even though previous publications showed that dietary spermidine extends longevity via induction of autophagy. I think the autophagy analyses should be strengthened to support these claims.

We thank the reviewer for recognizing the significance of our findings. We have followed the reviewer's recommendation and carried out additional autophagy analysis in both the *Drosophila* model and SRS patient cells. In particular, we added bone marrow stromal cells (BMSC), a clinically more relevant cell type, and performed additional autophagy analyses on two cell types (fibroblast and BMSC) from SRS patients. These additions have greatly strengthened the mechanistic analysis and further supported our findings.

Major comments

1) Fig 2c-e: are the *sms* mutant flies on a white+ background, or is the pale yellow coloration of the

eyes due to the transposon insertion in white- flies? If it is the latter case, then this experiment should be repeated in mutant animals with otherwise wild-type eye color, similar to controls shown in Fig 2a-b.

We thank the reviewer for pointing this out. The *dSms* mutant flies are on a *white-* background and the eye color is contributed by *mini-white+* transgene in the transposon element. We repeated the experiment as suggested with homozygous *UAS-dSms^{RA}* transgenic flies as controls for two reasons: first, this transgenic line has two copies of the *mini-white+* gene to match that of the homozygous *dSms* mutant flies in terms of eye color gene dosage; and second, this line is a parental control for *dSms^{RA}* overexpression in the rescue experiments (**Fig. 2f**).

We first reexamined the eye exterior morphology of the new control flies together with the *dSms* mutant flies (**Fig. 2a-e**) and observed pigmentation defects in the mutant flies (**Fig. 2c**) and retina degeneration with aging (**Fig. 2c-e**). Next we repeated the ERG analysis on the new control flies (**Fig. 2a', b'**) and re-quantified all the data (**Fig. 2g-h**). The new data are consistent with the results from previous experiments and support our original conclusions.

2) Cyto-ID-autophagy marker is seldom used for autophagy analysis, so its specificity is rather questionable (and I do not see that mutant cells take up less dye than controls upon chloroquine treatment in fig 3g, middle panels). I suggest that the authors repeat the tests shown in Fig 3g using Cherry-GFP-LC3B, the commonly used and specific marker of autophagosomes and autolysosomes. This would be important also because bafilomycin and rapamycin have no effect on LC3 lipidation in control cells (fig 3h).

We thank the reviewer for the suggestion and agree with the criticism. We followed the recommendation and used Cherry-GFP-LC3B to assess the autophagosomes and autolysosomes in SRS patient cells. Using the Premo Autophagy Tandem Sensor RFP-GFP-LC3B (Thermo Fisher Scientific), we found that the number of autophagosomes and autolysosomes were not significantly different between unaffected control and SRS fibroblasts (**Supplementary Fig. 5a, b**). Since we have previously found that SRS patient bone marrow stromal cells (BMSCs) exhibit a robust molecular phenotype with respect to polyamine levels (Albert et al., 2015), we repeated the analysis on BMSCs. Results of the RFP-GFP-LC3 assay show significantly increased numbers of autophagosomes and decreased numbers of autolysosomes in SRS BMSCs (**Fig. 4a, b**). In this revised version, we have included the results of RFP-GFP-LC3 analysis obtained from examining two cell types and discussed both results in the text.

To further support this finding, and to address the comment regarding the biochemical analysis of LC3 (original Fig. 3h), we repeated the Western analysis of LC3 on BMSCs. Consistent with other reports, we saw increased LC3-II levels in control cells when autophagy was either inhibited by bafilomycin or induced by starvation (**Fig. 4c, d**). In addition, lower levels of LC3-II were detected in SRS BMSCs compared with control group in all conditions (**Fig. 4c, d**), which suggest defects in LC3 turnover. However, we do not observe any significant differences between control and SRS fibroblasts in this assay. We have included both results and discussed them in the text.

3) A related comment is that the most common test for analyzing autophagic flux is by looking at the level of the specific autophagic cargo p62/SQSTM1 (Ref2P in flies). The level of endogenous p62 should be determined in control and mutant *Drosophila* and human cells, which will ideally support the suggested impairment of autophagic flux.

We followed the recommendation and determined Ref(2)P levels in flies and p62 levels in human BMSCs and fibroblasts. Using the Ref(2)P antibody reported (Pircs et al., 2012), we observed increased Ref(2)P positive puncta in the cell body layer of the lamina neurons in *dSms* mutant fly brains

(**Fig. 3d**). In human fibroblasts and BMSCs, we performed Western analysis for p62. We found that p62 levels increased upon autophagy inhibition and decreased upon autophagy induction as expected (**Fig. 4c, e, Supplementary Fig. 5e, f**). However, we observed no significant differences between control and SRS cells.

p62/Ref(2)p acts as cargo receptor specifically for degradation of ubiquitinated protein aggregates (Johansen and Lamark, 2011). The p62 protein level changes can be cell type and context specific (Discussed in (Klionsky et al., 2016)). The biochemical analysis *in vitro* may not be sensitive enough to detect the differences between control and SRS cells. We have included the p62 analysis results in the text.

4) Fig 4d: there is a huge difference in proenzyme levels between control samples, and one of the lots of bands is marked as mature CtsD. This blot is not convincing: more controls and a proof of antibody specificity are needed, or it should be left out.

We have repeated the biochemical analysis of CtsD with more controls, two of which are age matched with SRS patient cells (**Fig. 5h**). We observed consistent reduction in mature CtsD levels in SRS cells (**Fig. 5i**). In this experiment, we used the anti-cathepsin D antibody (R20, sc-6487, Santa Cruz) that has been verified and widely used in the literature (example references are listed in **Table R2**). Laurent-Matha et al. (Ref 1) have provided one of the most comprehensive analyses of human CtsD in fibroblasts. Our results are highly consistent with this study and a more recent study by Boonen et al (Ref 5) that examined the human CtsD in HeLa cells. To be consistent with these references, we labeled the two bands below 37kDa as mature form, the bands just below 50kDa as intermediates, and the band above 50kDa as pro form (52 kDa) of cathepsin D.

Table R2 Example references that examined mammalian Cathepsin D using the anti-CtsD antibody.

Reference	Tissue examined	Results
1. Laurent-Matha et al., 2006	Human fibroblasts	(Fig. 3) The study characterized the precursor (52kDa), processing intermediates (48kDa and 48-52kDa), as well as mature form (34kDa, and 34kDa with 9 amino acids) of human cathepsin D.
2. Sato et al., 2006	Rat cerebral cortex, hippocampus, lung, and spleen	(Fig. 5) Multiple bands were detected; patterns appeared to be tissue specific.
3. Vascotto et al., 2007	Mouse Spleen B cells	(Fig. 5D) Two bands at 32 (kDa) were detected as the mature form.
4. Kong et al., 2014	Mouse liver	(Fig. 7) Multiple bands were detected. Two bands were labeled as CtsD.
5. Boonen et al., 2016	Human HeLa cell line and a mouse neuroblastoma cell line (N1E-115)	(Fig. 4) The study detected precursor (band just above 50 KDa); intermediates (the band(s) between 37 and 50 kDa); and mature form (the band(s) below 37, labeled as heavy chain).

Minor comments:

5). Transposon insert seems to be in an intron in Fig 1b, unlike in the end of exon 3 as stated in the text.

We have corrected this to be “*with a transposable element inserted in the intron between exon 3 and 4*”.

6). Typo in Fig 1g: putrescien

We have corrected this to be “putrescine”.

7). Source and identity of Drosophila anti-CtsL antibody is missing from the Methods section.

We have added the source and identity in the Methods section.

Reviewer #3 (Remarks to the Author):

The manuscript by Li et al. describes a Drosophila model for the Snyder Robinson X-LID syndrome. In their experiments, they identify the generation of H₂O₂ and amine-containing aldehydes as key mechanistic features of the disease resulting in lysosomal dysfunction, in-line with previous data, and also mitochondrial dysfunction which represents a novel aspect of the disease. The manuscript offers a significant advance in the understanding of Snyder Robinson X-LID syndrome and key findings are supported by analysis of patient cells. The experiments are largely (with some exceptions, see below) convincing and well done. A few aspects need to be addressed, however, before the manuscript is appropriate for publication.

We thank the reviewer for the enthusiastic and supportive comments. We have addressed each comment with additional experiments and as a result the manuscript is greatly improved and strengthened.

1) Several of the key experiments assessing Drosophila phenotypes are shown without rescue controls, despite those flies being available. While this may be reasonable for measurements of metabolites directly related to SMS activity (like spermidine and putrescin (in Figure 1) and their metabolites (in Figure 4A,B), it is not acceptable for the measurement of complex phenotypes (like longevity, negative geotaxis and neurodegeneration) which are easily influenced by different genetic backgrounds and require rescue experiments to unequivocally assign phenotypes to specific genes. As the necessary flies are available (See figure 1e) and the assays established, this should not be a significant problem delaying the manuscript.

We agree with the reviewer on the important issue of controls. We overexpressed dSms isoform dSms^{RA} in homozygous dSms mutant flies and carried out a series of experiments to examine the fly phenotypes with these rescue controls. (1) Life span and behavior: dSms^{RA} overexpression significantly restored the locomotor behavior (**Fig. 1h**) and significantly rescued the shortened life span of dSms mutant flies (**Fig. 1i**). (2) Neurodegeneration phenotypes: we examined the function and morphology of the visual system. dSms^{RA} overexpression significantly rescued the retina degeneration (**Fig. 2f**) and synaptic function measured by ERG recording at DAE30 (**Fig. 2f, g-i**). (3) Mitochondria function: dSms^{RA} overexpression restored COX activity on DAE5 in fly brains and flight muscles (**Fig. 6e**).

2) The claimed “enrichment” of SMS at synapses is not convincingly demonstrated. First, there are no data provided which show that the GFP trap line encodes a functional fusion protein, whose localization reflects that of the endogenous protein. Second, the low-resolution images provided are just as consistent with distribution throughout cells as they are with enrichment at the Brp-labeled synapses.

This claim needs to be better supported or could be removed without reducing the impact of the manuscript.

We agree with the reviewer that because it is unclear whether the GFP trap line encodes a functional protein, the GFP signal only reflects a cell type specific expression, but not a compartmentalized localization of dSms protein.

Although we do not have the antibody to examine the endogenous localization of dSms, we addressed this question with an alternative approach. We overexpressed HA tagged dSms^{RA} in *dSms* mutant background using *actin-gal4* driver and examined protein localization. The HA-dSms^{RA} is functional, as it can rescue viability, lifespan, locomotor behavior, and neurodegeneration of mutant flies. We performed immunofluorescent staining in the lamina synapses. We labeled neuronal membranes with HRP and synapses with CSP. HA-dSms^{RA} is present in the synapses.

However, we decided not to include this figure in the manuscript and deleted the description regarding 'synaptic enrichment' for two reasons: 1) dSms^{RA} is ectopically expressed, therefore may not be the same as endogenous localization; and 2) dSms^{RA} is one of the two predicted isoform of dSms and may not truly reflect all isoforms that are endogenously expressed.

3) The effects of Bafilomycin and Rapamycin on the LC3-I/-II ratios in Figure 3 h are not obvious and need to be quantified.

We repeated the biochemical analysis of LC3 on BMSCs and observed results consistent with previous reports. Specifically, we saw increased LC3-II levels in control cells when autophagy was either inhibited by bafilomycin or induced by starvation (**Fig. 4c, d**). In addition, lower levels of LC3-II were detected in SRS BMSCs compared with control group in all conditions, which suggest defects in LC3 turnover (**Fig. 4c, d**). We added quantification in the revised version (**Fig. 4d**). However, we do not observe any significant differences between control and SRS fibroblasts in this assay. We have included both results in the text.

4) The authors demonstrate a remarkable reduction of pathogenic ROS production by supplementing food with AD4 or N-2-MGP. Very disappointingly they do not show the phenotypic consequences on the tissue distribution of Cox activity (as shown in Fig. 5e) neurodegeneration or behavioral outcomes.

[Redacted]

Even if the outcomes are “negative”, these would be important additions significantly strengthening the manuscript.

We agree with the reviewer that regardless of the outcome, the analysis is important to include to strengthen the study. To address this point and a similar point raised by Reviewer 1 (see above), we carried out additional experiments to comprehensively examine the effects of antioxidants by genetic manipulation and pharmacological administration on *dSms* mutant phenotypes.

Genetic manipulation: We overexpressed GstE1 in homozygous *dSms* mutant flies and found that 1) GstE1 overexpression significantly extended life span of *dSms* mutant flies (**Fig. 7d**); 2) GstE1 overexpression significantly rescued the pigmentation changes at DAE15 and DAE30 (**Supplementary Fig. 9**) and the retina physiology measured by ERG at DAE15 and DAE30 (**Fig. 7e-h**); and 3) GstE1 overexpression restored COX activity at DAE5, especially in fly flight muscles (**Fig. 7k, Supplementary Fig. 10**).

Pharmacological administration: We supplemented the food with antioxidant compounds at different concentrations and found that, in addition to reducing brain ROS accumulation, AD4 feeding increased COX activity suggesting partial restoration of mitochondria function (**Fig. 7k, Supplementary Fig. 10**). However, administration of AD4 (40 µg/ml) or N-2-MPG (160 µg/ml) did not restore lysosome integrity as measured by LAMP1 and cathepsin L immune-labeling (**Supplementary Fig. 11**). This is not surprising as lysosome defects are likely caused by aldehyde accumulation that is not targeted by these anti-oxidant compounds. We also examined the effects of antioxidant feeding on survival. *dSms* mutant flies have a reduced survival rate, that is less than 50% of the flies complete metamorphosis and eclosion process. Feeding *dSms* mutant larvae with antioxidant AD4 (40 µg/ml) did not significantly improve viability (**Sup. Fig. 8**), suggesting that feeding for three days at larval stage is insufficient to overcome metamorphosis defects and improve eclosion rate.

References

- Albert, J.S., Bhattacharyya, N., Wolfe, L.A., Bone, W.P., Maduro, V., Accardi, J., Adams, D.R., Schwartz, C.E., Norris, J., Wood, T., *et al.* (2015). Impaired osteoblast and osteoclast function characterize the osteoporosis of Snyder - Robinson syndrome. *Orphanet J Rare Dis* **10**, 27.
- Boonen, M., Staudt, C., Gilis, F., Oorschot, V., Klumperman, J., and Jadot, M. (2016). Cathepsin D and its newly identified transport receptor SEZ6L2 can modulate neurite outgrowth. *J Cell Sci* **129**, 557-568.
- Eisenberg, T., Knauer, H., Schauer, A., Buttner, S., Ruckenstuhl, C., Carmona-Gutierrez, D., Ring, J., Schroeder, S., Magnes, C., Antonacci, L., *et al.* (2009). Induction of autophagy by spermidine promotes longevity. *Nat Cell Biol* **11**, 1305-1314.
- Gupta, V.K., Scheunemann, L., Eisenberg, T., Mertel, S., Bhukel, A., Koemans, T.S., Kramer, J.M., Liu, K.S., Schroeder, S., Stunnenberg, H.G., *et al.* (2013). Restoring polyamines protects from age-induced memory impairment in an autophagy-dependent manner. *Nat Neurosci* **16**, 1453-1460.
- Ha, H.C., Sirisoma, N.S., Kuppusamy, P., Zweier, J.L., Woster, P.M., and Casero, R.A., Jr. (1998). The natural polyamine spermine functions directly as a free radical scavenger. *Proc Natl Acad Sci U S A* **95**, 11140-11145.
- Johansen, T., and Lamark, T. (2011). Selective autophagy mediated by autophagic adapter proteins. *Autophagy* **7**, 279-296.
- Klionsky, D.J., Abdelmohsen, K., Abe, A., Abedin, M.J., Abeliovich, H., Acevedo Arozena, A., Adachi, H., Adams, C.M., Adams, P.D., Adeli, K., *et al.* (2016). Guidelines for the use and interpretation of assays for monitoring autophagy (3rd edition). *Autophagy* **12**, 1-222.

Kong, X.Y., Nasset, C.K., Damme, M., Loberg, E.M., Lubke, T., Maehlen, J., Andersson, K.B., Lorenzo, P.I., Roos, N., Thoresen, G.H., *et al.* (2014). Loss of lysosomal membrane protein NCU-G1 in mice results in spontaneous liver fibrosis with accumulation of lipofuscin and iron in Kupffer cells. *Dis Model Mech* 7, 351-362.

Laurent-Matha, V., Derocq, D., Prebois, C., Katunuma, N., and Liaudet-Coopman, E. (2006). Processing of human cathepsin D is independent of its catalytic function and auto-activation: involvement of cathepsins L and B. *J Biochem* 139, 363-371.

Merkulova, M., Paunescu, T.G., Azroyan, A., Marshansky, V., Breton, S., and Brown, D. (2015). Mapping the H(+) (V)-ATPase interactome: identification of proteins involved in trafficking, folding, assembly and phosphorylation. *Sci Rep* 5, 14827.

Minois, N., Rockenfeller, P., Smith, T.K., and Carmona-Gutierrez, D. (2014). Spermidine feeding decreases age-related locomotor activity loss and induces changes in lipid composition. *PLoS One* 9, e102435.

Pavel, M., Imarisio, S., Menzies, F.M., Jimenez-Sanchez, M., Siddiqi, F.H., Wu, X., Renna, M., O'Kane, C.J., Crowther, D.C., and Rubinsztein, D.C. (2016). CCT complex restricts neuropathogenic protein aggregation via autophagy. *Nat Commun* 7, 13821.

Pircs, K., Nagy, P., Varga, A., Venkei, Z., Erdi, B., Hegedus, K., and Juhasz, G. (2012). Advantages and limitations of different p62-based assays for estimating autophagic activity in *Drosophila*. *PLoS One* 7, e44214.

Sato, Y., Suzuki, Y., Ito, E., Shimazaki, S., Ishida, M., Yamamoto, T., Yamamoto, H., Toda, T., Suzuki, M., Suzuki, A., *et al.* (2006). Identification and characterization of an increased glycoprotein in aging: age-associated translocation of cathepsin D. *Mech Ageing Dev* 127, 771-778.

Vascotto, F., Lankar, D., Faure-Andre, G., Vargas, P., Diaz, J., Le Roux, D., Yuseff, M.I., Sibarita, J.B., Boes, M., Raposo, G., *et al.* (2007). The actin-based motor protein myosin II regulates MHC class II trafficking and BCR-driven antigen presentation. *J Cell Biol* 176, 1007-1019.

Reviewers' comments:

Reviewer #1 (Remarks to the Author):

The authors have addressed my concerns, and the paper is ready for acceptance.

Reviewer #2 (Remarks to the Author):

In their revised manuscript, Li et al include most of the requested experiments, but a couple of questions are left open, especially regarding the relevance of their findings to patient neurons. I list here my most important concerns:

1. There seems to be a problem with lysosome acidification (Fig 5J, K), which probably accounts for the autophagy phenotypes. Since lysosome acidification is not required for fusion with autophagosomes (Nat Commun. 2015 May 11;6:7007.), structures positive for both GFP and RFP using the tandem tagged reporter probably represent dysfunctional autolysosomes rather than autophagosomes. Interpretation of these data should be updated accordingly.
2. The autophagy defect is variable between cell types (SRS fibroblasts vs BMSCs) and tests (Atg8/LC3 vs p62), so it is really not clear whether there is a defect in patient neurons. This is a clear weakness of the present study, which could be addressed by somehow testing neurons (these could also be generated from other types of cells derived from patients).
3. Eye colour of miniwhite+ transposon-bearing flies depends heavily on chromosomal insertion effects, so Fig2a,b is not a good control for Fig 2c-f. dSMS is located on chromosome 3, so it would be easy to cross in a wild type white+ bearing X chromosome into these backgrounds and repeat the eye colour tests.

Reviewer #3 (Remarks to the Author):

I have reviewed the revised manuscript and response to the previous critiques. The authors have now addressed my main concerns and I believe that the manuscript is now acceptable for publication in Nature Communications.

Point-to-point response to reviewers' comments:

Reviewer #1 (Remarks to the Author):

The authors have addressed my concerns, and the paper is ready for acceptance.

We thank the reviewer for the recommendation to publish our work in *Nature Communications*!

Reviewer #2 (Remarks to the Author):

In their revised manuscript, Li et al include most of the requested experiments, but a couple of questions are left open, especially regarding the relevance of their findings to patient neurons. I list here my most important concerns:

1. There seems to be a problem with lysosome acidification (Fig 5J, K), which probably accounts for the autophagy phenotypes. Since lysosome acidification is not required for fusion with autophagosomes (Nat Commun. 2015 May 11;6:7007.), structures positive for both GFP and RFP using the tandem tagged reporter probably represent dysfunctional autolysosomes rather than autophagosomes. Interpretation of these data should be updated accordingly.

We agree with the reviewer that lysosomal acidification is not required for fusion with autophagosomes, thus the puncta positive for both GFP and RFP signals can be autophagosomes or dysfunctional autolysosomes with a neutral pH. The defect in lysosome acidification is also supported by our biochemical analysis of cathepsins (Fig. 5e, f, h, i), since the acidic environment is required for the lysosomal process of cathepsins. We have modified our interpretation of the mCherry -GFP-Atg8a assay in flies and the RFP-GFP-LC3 assay in human cells.

The revised text reads, on page 7-8,

'Puncta with high mCherry and high GFP signals (Fig. 3h, quadrant I) indicate either Atg8a positive autophagosomes or dysfunctional autolysosomes with a neutral pH, since fusion of autophagosome and lysosome can proceed independent of lysosome acidification (Mauvezin et al., 2015). Puncta with high mCherry and low GFP signals (Fig. 3h, quadrant II) indicate fusion with the acidic lysosome. Puncta with low mCherry and low GFP signals (Fig. 3h, quadrant III) indicate normal lysosomal degradation of Atg8a. With this classification system we found that within the control lamina 25% of Atg8 puncta labeled autophagosomes (I), about 25% labeled autophagosomes undergoing fusion with the lysosome (II), and more than 50% were undergoing degradation within the lysosome (III); this defined basal autophagic flux (Fig. 3f, h). dSms^{el} lamina, however, had significantly more Atg8 puncta associated with autophagosomes or dysfunctional autolysosomes (I) and fewer puncta incorporated into normal autolysosomes (II and III) as well as increased GFP intensity throughout the lamina (Fig. 3f-h and Supplementary Fig. 4b).'

'We observed no difference in autophagic flux in fibroblasts from SRS patients, however we observed significantly more autophagosomes or dysfunctional autolysosomes along with fewer acidic autolysosomes in bone marrow stromal cells (BMSCs) from SRS patients, a cell type which exhibits a robust molecular phenotype with respect to polyamine levels compared to SRS fibroblasts (Albert et al., 2015) (Fig. 4a, b, Supplementary Fig. 5a, b).'

2. The autophagy defect is variable between cell types (SRS fibroblasts vs BMSCs) and tests (Atg8/LC3 vs p62), so it is really not clear whether there is a defect in patient neurons. This is a clear weakness of the present study, which could be addressed by somehow testing neurons (these could also be generated from other types of cells derived from patients).

We would like to first emphasize that SRS is a multi-system disorder most severely affecting nervous system, and skeletal tissues (Albert et al., 2015; Becerra-Solano et al., 2009; Cason et al., 2003; de Alencastro et al., 2008; Zhang et al., 2013). SMS is widely expressed, however it is expected that different cells will have varied tolerance of SMS deficiency. Previous studies compared bone-derived BMSCs, lymphoblastoid cells, and skin-derived fibroblasts, and found that although the SMS mRNA and protein expression levels were similar, the cellular polyamine content was much more skewed in BMSCs than that in lymphoblasts or fibroblasts (Albert et al., 2015). As lymphoblastoid cells and skin fibroblasts are less affected tissues compared to bone in SRS, the severity of polyamine imbalance is correlated with SRS pathogenesis (Albert et al., 2015; Becerra-Solano et al., 2009; de Alencastro et al., 2008). To examine cell-type specific phenotypes, we included three types of patient cells in this study, fibroblasts (Fig 5, Supplementary Fig. 5, 6), BMSCs (Fig 4), and lymphoblastoid cells (Fig. 6).

Our biochemical analysis of autophagy showed a significantly different extent of autophagic response assessed by LC3-II (Fig. 4c-e) in SRS patient-derived BMSCs, but not in SRS fibroblasts (Supplementary Fig. 5). However, analysis on p62 level did not show a robust phenotype (Fig. 4c-e). We noted that p62 is a marker for ubiquitinated protein aggregates degraded through autophagy (Pircs et al., 2012), and p62 changes can be cell type and context specific (Klionsky et al., 2016). These results together with our findings in *Drosophila* brain support our model that polyamine imbalance and cellular defects explains the tissue-specific manifestation in SRS.

Although it would be ideal to examine patient neurons, these samples are unfortunately not available, nor are postmortem tissues. In recent years, cultured neuronal cells differentiated from reprogrammed iPSCs have been established for several neurological disorders (Mertens et al., 2016). Studies using these cells have provided insights into aspects of disease mechanisms (Ichida and Kiskinis, 2015). So far in SRS research community, iPSCs have not been successfully established. Establishing the iPSCs, optimizing their differentiation into neurons, and characterizing the neuronal cells when established are far beyond the scope of the current study and paper. In this study, we use *in vivo* analyses in *Drosophila* nervous system to overcome the difficulty and limitation of patient-derived neuronal cells, and use established SRS patient cell lines to demonstrate conservation of cellular mechanisms. We show that loss of *dSms* causes lysosome dysfunction, and impairs autophagic flux in the nervous system *in vivo* in *Drosophila*. This cellular mechanism is confirmed in patient BMSCs, derived from highly affected patient tissue. Such parallel observations in different cell types and tissues have two important implications: first, different cell types have varied tolerance to polyamine imbalance; and second, autophagy and other cellular phenotypes are correlated with polyamine phenotypes and SRS disease pathophysiology. Combining investigations into SMS-deficient *Drosophila* nervous system with SRS patient cells (fibroblasts, lymphoblasts, and BMSCs) is one of the strengths of our approach that allowed us to uncover mechanisms that underlie manifestations in different systems in SRS.

We have highlighted these points in the text (Line 175-180, 190-193, 196-197), and added a new paragraph in the discussion (Page 17).

3. Eye colour of miniwhite+ transposon-bearing flies depends heavily on chromosomal insertion effects, so Fig2a,b is not a good control for Fig 2c-f. dSMS is located on chromosome 3, so it would be easy to cross in a wild type white+ bearing X chromosome into these backgrounds and repeat the eye colour tests.

We agree with the reviewer that the eye color can be affected by chromosomal insertion effects. However, we would like to clarify that the primary phenotype in mutant flies was not eye color per se, but rather the abnormal age-dependent loss of pigmentation, a hallmark for retinal degeneration (Li et al., 2008). The retinal abnormality is further suggested by abnormal retinal vacuoles observed in 2 DAE mutant flies (Supplementary Fig. 3). These observations led us to test the phototransduction using ERG analysis, and we observed age-dependent reduction in phototransduction and synaptic transmission. Our characterization of SMS deficiency in *Drosophila* was carried out in females so crossing into a wild type *white+* bearing X chromosome runs a risk of highly overexpressing the *white* gene in SMS homozygous mutant females (2 copies of *white+* X chromosome plus two copies of *miniwhite+*), potentially masking or delaying the onset of the pigmentation defect. For this reason, we considered similar *miniwhite+* transposon-bearing flies in a *white-* background as the optimal control for analysis of retinal degeneration phenotypes.

We acknowledge the potential confusion that our description may have caused, and have revised the text to clarify this point and emphasize the pigmentation defects.

The revised text reads, on page 6,

'To establish the nervous system requirement of dSms, we focused on the visual system where we found abnormal retinal vacuoles in dSms^{e/e} flies at 2 days after eclosion (DAE) (Supplementary Fig. 3). Additionally, we observed an age-dependent (Fig. 2c, d, e) loss of pigmentation in dSms^{e/e} flies characteristic of retinal degeneration (Li et al., 2008).'

Reviewer #3 (Remarks to the Author):

I have reviewed the revised manuscript and response to the previous critiques. The authors have now addressed my main concerns and I believe that the manuscript is now acceptable for publication in Nature Communications.

We thank the reviewer for the recommendation to publish our work in *Nature Communications!*

References

- Albert, J.S., Bhattacharyya, N., Wolfe, L.A., Bone, W.P., Maduro, V., Accardi, J., Adams, D.R., Schwartz, C.E., Norris, J., Wood, T., *et al.* (2015). Impaired osteoblast and osteoclast function characterize the osteoporosis of Snyder - Robinson syndrome. *Orphanet J Rare Dis* 10, 27.
- Becerra-Solano, L.E., Butler, J., Castaneda-Cisneros, G., McCloskey, D.E., Wang, X., Pegg, A.E., Schwartz, C.E., Sanchez-Corona, J., and Garcia-Ortiz, J.E. (2009). A missense mutation, p.V132G, in the X-linked spermine synthase gene (SMS) causes Snyder-Robinson syndrome. *Am J Med Genet A* 149A, 328-335.
- Cason, A.L., Ikeguchi, Y., Skinner, C., Wood, T.C., Holden, K.R., Lubs, H.A., Martinez, F., Simensen, R.J., Stevenson, R.E., Pegg, A.E., *et al.* (2003). X-linked spermine synthase gene (SMS) defect: the first polyamine deficiency syndrome. *Eur J Hum Genet* 11, 937-944.
- de Alencastro, G., McCloskey, D.E., Kliemann, S.E., Maranduba, C.M., Pegg, A.E., Wang, X., Bertola, D.R., Schwartz, C.E., Passos-Bueno, M.R., and Sertie, A.L. (2008). New SMS mutation leads to a striking reduction in spermine synthase protein function and a severe form of Snyder-Robinson X-linked recessive mental retardation syndrome. *J Med Genet* 45, 539-543.

Ichida, J.K., and Kiskinis, E. (2015). Probing disorders of the nervous system using reprogramming approaches. *EMBO J* 34, 1456-1477.

Klionsky, D.J., Abdelmohsen, K., Abe, A., Abedin, M.J., Abeliovich, H., Acevedo Arozena, A., Adachi, H., Adams, C.M., Adams, P.D., Adeli, K., *et al.* (2016). Guidelines for the use and interpretation of assays for monitoring autophagy (3rd edition). *Autophagy* 12, 1-222.

Li, L.B., Yu, Z., Teng, X., and Bonini, N.M. (2008). RNA toxicity is a component of ataxin-3 degeneration in *Drosophila*. *Nature* 453, 1107-1111.

Mauvezin, C., Nagy, P., Juhasz, G., and Neufeld, T.P. (2015). Autophagosome-lysosome fusion is independent of V-ATPase-mediated acidification. *Nat Commun* 6, 7007.

Mertens, J., Marchetto, M.C., Bardy, C., and Gage, F.H. (2016). Evaluating cell reprogramming, differentiation and conversion technologies in neuroscience. *Nat Rev Neurosci* 17, 424-437.

Pircs, K., Nagy, P., Varga, A., Venkei, Z., Erdi, B., Hegedus, K., and Juhasz, G. (2012). Advantages and limitations of different p62-based assays for estimating autophagic activity in *Drosophila*. *PLoS One* 7, e44214.

Zhang, Z., Norris, J., Kalscheuer, V., Wood, T., Wang, L., Schwartz, C., Alexov, E., and Van Esch, H. (2013). A Y328C missense mutation in spermine synthase causes a mild form of Snyder-Robinson syndrome. *Hum Mol Genet* 22, 3789-3797.

REVIEWERS' COMMENTS:

Reviewer #2 (Remarks to the Author):

I think that with the added clarifications and discussions, this manuscript is now acceptable for publication.